# SurvivalPFN: Prior-Data Fitted Networks for Survival Analysis

## Abstract

Survival analysis models time-to-event outcomes and is widely used across many domains. However, real-world cohorts are often small and heavily censored, where deep models are prone to instability while classical methods remain strong baselines. We propose SurvivalPFN, a Transformer that follows the Prior-data Fitted Networks (PFNs) paradigm to perform amortized Bayesian survival inference. Pretrained once on synthetic survival tasks drawn from broad priors, SurvivalPFN learns to map a labeled training set directly to a dataset-conditional survival function. At inference time, it consumes the training split of the target dataset as in-context input and produces the corresponding survival function in a single forward pass, without per-dataset gradient updates. Experiments on real-world survival benchmarks show that SurvivalPFN is competitive with strong baselines, demonstrating that prior-driven in-context prediction is an effective approach for small-sample survival analysis.

## 1. Introduction

Survival analysis studies time-to-event outcomes under censoring; its core summaries are the survival function $S(t \mid x) = \mathbb{P}(T > t \mid x)$ and the hazard rate $h(t \mid x)$, which characterize how event-time distributions evolve over time (Kaplan & Meier, 1958; Cox, 1972). It underpins clinical prognosis (e.g., cancer), reliability engineering (e.g., machine failure), and risk modeling such as customer churn or credit default (Altman & Bland, 1998; Nelson, 1972; Banasik et al., 1999). In many applications, datasets are *small* and heavily censored: long follow-up, privacy constraints, and rare events limit data collection, while censoring further reduces effective information. In medical applications in particular, privacy regulations (e.g., the GDPR (Parliament & the Council of the European Union, 2016)) constrain data sharing, and limited automation in data collection and curation can make research-ready clinical datasets costly to build (de Kok et al., 2023). Moreover, some clinically informative covariates arise from invasive procedures that add patient burden and risk (Dermody & Shuman, 2021). Small sample sizes also make model selection and validation noisy, motivating methods that can reuse statistical structure across datasets. Many downstream decisions require calibrated survival curves (not only risk ranking), further increasing the burden in small cohorts.

Classical approaches such as Cox proportional hazards remain efficient and interpretable in these settings (Cox, 1972). Under the proportional hazards assumption (time-invariant hazard ratios), they provide well-understood statistical properties, but the log-risk is modeled linearly and complex interactions can be missed. To capture nonlinear effects with minimal modeling assumptions, nonparametric ensembles such as Random Survival Forests can remain hard to beat in limited-event cohorts; for example, Random Survival Forests outperformed Cox regression-based and other ML models for post-transplant survival prediction in myelofibrosis (Hernández-Boluda et al., 2025). Deep learning methods such as DeepSurv (Katzman et al., 2018) increase flexibility, yet in small-sample regimes they are prone to overfitting and require per-dataset tuning; validation becomes unreliable when event counts are low and censoring is heavy (Wiegrebe et al., 2024).

To reuse statistical structure across datasets without per-dataset optimization, a recent line of work on small-data learning, Prior-Data Fitted Networks (PFNs), aims to amortize Bayesian inference across datasets: a Transformer is trained on many synthetic datasets drawn from an explicit task prior, amortizing posterior prediction into its weights (Müller et al., 2021; Hollmann et al., 2023; Nagler, 2023). As a result, inference on a new dataset reduces to a single forward pass conditioned on its labeled training split, avoiding per-dataset gradient updates and hyperparameter tuning. In this framing, each dataset is a *task*: its observed covariates and censored times form the context, and we aim to predict survival for new covariates from the same dataset. Formally, given a context set $D_C = \{(x_i, Z_i, \delta_i)\}_{i=1}^n$ and a query covariate $x^\star$, we seek a dataset-conditioned survival

[1]Anonymous Institution, Anonymous City, Anonymous Region, Anonymous Country. Correspondence to: Anonymous Author <anon.email@domain.com>.

Preliminary work. Under review by the International Conference on Machine Learning (ICML). Do not distribute.

function

$$S(t \mid x^\star, D_C) \;=\; \mathbb{P}(T^\star > t \mid x^\star, D_C), \qquad (1)$$

where $Z_i$ is the observed time and $\delta_i$ indicates whether the event was observed.

We propose *SurvivalPFN*, a Transformer in the Prior-Data Fitted Networks paradigm for amortized survival inference. SurvivalPFN is trained solely on synthetic censored survival tasks sampled from a broad prior over hazard mechanisms, time scales, covariate effects, and censoring regimes, using a censoring-consistent likelihood. At inference time, the available training split serves as labeled context and SurvivalPFN outputs survival functions for the test split in a single forward pass, without gradient-based fine-tuning.

**Contributions.** (1) To the best of our knowledge, we introduce the first Prior-Data Fitted Network for censoring-aware survival prediction trained solely on synthetic censored survival tasks, enabling dataset-conditioned inference by conditioning on the training split as context without per-dataset optimization. (2) We design a broad synthetic survival-task prior with diverse hazard mechanisms, time scales, covariate effects, and censoring patterns; sampling from this prior provides the synthetic tasks for training and supports amortized downstream inference. (3) On real-world survival benchmarks, SurvivalPFN achieves the best dataset-equal macro averages across C-index, iAUC, and IBS in our benchmark, while requiring inference only on downstream datasets. (4) Code and pretrained weights are available at `https://anonymous.4open.science/r/SurvivalPFN`.

## 2. Related Work

### 2.1. Survival models

Survival analysis concerns time-to-event outcomes observed under censoring. Most methods assume non-informative censoring and model the event-time distribution through the survival function $S(t \mid x)$ or hazard $h(t \mid x)$. Classical regression families remain widely used: Cox proportional hazards models covariates as a time-invariant multiplicative effect on the hazard while leaving the baseline unspecified (Cox, 1972), whereas accelerated failure-time models represent covariates as a multiplicative time-scaling of the event-time distribution (often linear in $\log T$) (Wei, 1992). Nonparametric estimators such as Kaplan–Meier (Kaplan & Meier, 1958) and counting-process formulations of cumulative hazards (Aalen, 1978) provide foundational tools and are commonly used for evaluation and calibration.

To capture nonlinear effects with minimal modeling assumptions, tree-based ensembles such as Random Survival Forests (Ishwaran et al., 2008) and survival boosting (Binder & Schumacher, 2008) are popular and serve as strong default baselines, particularly in small cohorts with limited event counts.

Deep survival models replace linear predictors with neural networks while retaining censoring-aware objectives. Cox-style approaches parameterize the log-risk with a neural network (DeepSurv (Katzman et al., 2018), Cox-nnet (Ching et al., 2018)) and extend to time-varying effects (Cox-Time/CoxCC (Kvamme et al., 2019)); other methods directly parameterize the event-time distribution, e.g., mixture-based Deep Survival Machines (Nagpal et al., 2021). Transformers have been explored for longitudinal covariates (Hu et al., 2021; Wang & Sun, 2022) and sequence-level event modeling (Öğretir et al., 2024; Zisser & Aran, 2024), and representation-learning objectives can improve discrimination under censoring (Lee et al., 2024). In practice, however, deep models can be sensitive to optimization and per-dataset tuning in small or heavily censored cohorts (Wiegrebe et al., 2024).

Discrete-time survival modeling discretizes follow-up into bins and predicts either per-interval hazards (yielding a product-form survival curve) or categorical event-time mass over bins. Hazard-based parameterizations yield monotone survival by construction and admit straightforward censoring-consistent likelihoods with flexible baseline hazards (Gensheimer & Narasimhan, 2019). Mass-based approaches instead predict event probabilities over bins as in DeepHit (Lee et al., 2018), while MTLR provides an alternative discretized formulation via dependent regressors over bins (Yu et al., 2011). Binning choices and horizon truncation affect data sparsity and calibration, especially under heavy censoring (Wiegrebe et al., 2024).

Finally, beyond the non-informative censoring assumption, recent work explicitly models censoring mechanisms and informative censoring (Shahin et al., 2024).

### 2.2. Prior-Data Fitted Networks and in-context learning

Prior-Data Fitted Networks (PFNs) can be viewed as in-context predictors: they train Transformers to predict a query target by conditioning on a small labeled dataset provided as context (Müller et al., 2021; Hollmann et al., 2023; Nagler, 2023). Formally, the model learns a set-to-point mapping from a labeled context set $D_C = \{(x_i, y_i)\}_{i=1}^n$ and a query input $x^\star$ to a predictive distribution $p(y^\star \mid x^\star, D_C)$, typically implemented by attending over the context examples. PFNs are pretrained on many synthetic datasets sampled from an explicit task prior; under a latent-task model with parameters $\theta \sim p(\theta)$ and data generation $p(y \mid x, \theta)$, the prior induces the Bayesian posterior predictive distribution

$$p(y^\star \mid x^\star, D_C) = \int p(y^\star \mid x^\star, \theta)\, p(\theta \mid D_C)\, \mathrm{d}\theta, \qquad (2)$$

which PFNs aim to approximate amortizedly, turning posterior prediction into a single forward pass at inference time. At inference time, the labeled training split of a new dataset is provided as context and predictions are produced without gradient-based adaptation.

TabPFN demonstrates this paradigm for small-data classification (Hollmann et al., 2023). Subsequent work establishes statistical foundations (Nagler, 2023), improves scalability and robustness via sketching/feature selection and context optimization (Feuer et al., 2023; 2024), and studies uncertainty quantification (Nagler & Rügamer, 2025). PFN-style synthetic pretraining has been extended beyond classification, including zero-shot forecasting (Dooley et al., 2023) and in-context causal effect estimation (Robertson et al., 2025; Balazadeh et al., 2025). Together, these directions advance PFNs along three axes: statistical understanding, scalability, and uncertainty. Related directions investigate context selection and distillation for vector-valued prediction tasks (Ma et al., 2024; Qu et al., 2025).

## 3. Method

We consider a dataset consisting of observations $\{(x_i, Z_i, \delta_i)\}_{i=1}^n$, where $x_i \in \mathbb{R}^d$, $Z_i = \min(T_i, U_i)$ is the observed time with event time $T_i$ and censoring time $U_i$, and $\delta_i = \mathbb{I}[T_i \leq U_i] \in \{0, 1\}$ indicates whether the event is observed. Let $D_C = \{(x_i, Z_i, \delta_i)\}_{i \in C}$ denote a labeled context set and $D_Q = \{(x_i, Z_i, \delta_i)\}_{i \in Q}$ query points from the same dataset. SurvivalPFN predicts a dataset-conditioned survival function $S(t \mid x, D_C) = \mathbb{P}(T > t \mid x, D_C)$ for query samples by attending to the context set $D_C$. Specifically, we normalize time per dataset using the context set and discretize it onto a fixed grid to align heterogeneous time scales. We then parameterize survival via nonnegative discrete-time integrated-hazard increments, which guarantees monotone survival functions by construction, and train with a censoring-consistent likelihood. Finally, we pretrain on synthetic survival tasks sampled from a designed prior, enabling PFN-style in-context prediction at inference time.

### 3.1. Time normalization and discretization

To align heterogeneous time scales across datasets, we discretize time on a fixed grid *after* dataset-specific normalization derived from the context set. This produces a normalized time coordinate $u \in [0, 1]$, enabling a shared discretization with fixed resolution $J$ across datasets. All dataset-specific time-normalization statistics (including $\kappa$) are computed from the labeled context set $D_C$; at inference, we use the dataset's training split as context. Let $z \geq 0$ denote the observed time and apply the stabilizing transform

$$s = \log(1 + z). \tag{3}$$

We estimate a dataset-specific time scale from the context times $\{s_i\}_{i \in C}$ via

$$\kappa = \text{Quantile}_q(\{s_i\}_{i \in C}) + \varepsilon_t, \tag{4}$$

with fixed $q \in (0, 1]$ and a small log-time offset $\varepsilon_t$ (we use $\varepsilon_t = 1e - 8$) for numerical stability. We then normalize and clip

$$u = \text{clip}(s/\kappa, 0, 1) \in [0, 1]. \tag{5}$$

Clipping at $u = 1$ imposes a finite prediction horizon in raw time, $t_{\text{hor}} = \exp(\kappa) - 1$. We therefore formulate learning and evaluation on $[0, t_{\text{hor}}]$ by recoding observations as $(z, \delta) \mapsto (\min(z, t_{\text{hor}}), \delta \mathbf{1}[z \leq t_{\text{hor}}])$, so that $\delta$ indicates events observed within the horizon. We apply this recoding to both context and query points and reuse $(z, \delta)$ notation below.

We use $J$ right-closed bins in raw time with endpoints $0 = t_0 < t_1 < \cdots < t_J$, defined by a uniform grid in the transformed domain:

$$s_j = \frac{j}{J}\kappa, \qquad t_j = \exp(s_j) - 1, \qquad j = 0, 1, \ldots, J. \tag{6}$$

Note that $t_J = t_{\text{hor}}$. Uniform spacing in $s$ yields approximately logarithmic spacing in raw time, allocating finer resolution at early times. Each bin is $(t_y, t_{y+1}]$ for $y = 0, \ldots, J - 1$. Given an observation with normalized time $u$, we assign it to the smallest bin end whose normalized location is at least $u$:

$$y = \min\big(\max(0, \lceil uJ \rceil - 1), J - 1\big) \\ \in \{0, \ldots, J - 1\}. \tag{7}$$

(Convention: $u = 0$ is assigned to $y = 0$.) Label $y$ corresponds to interval $(t_y, t_{y+1}]$ and bin end $t_{y+1}$. In particular, if the pre-truncation observed time exceeds $t_{\text{hor}}$, then after truncation $z = t_{\text{hor}}$, $\delta = 0$, $u = 1$, and $y = J - 1$.

For the PFN input, we represent the time label as a normalized bin index $\tilde{y} = y/(J - 1) \in [0, 1]$ together with the event indicator $\delta$. The model returns survival values at the $J$ nonzero bin ends $\{t_1, \ldots, t_J\}$ (with $t_0 = 0$ implicit), yielding a right-continuous survival curve in the original time units.

### 3.2. Discrete-time integrated hazard parameterization

We parameterize survival via nonnegative *integrated hazard increments* on the normalized grid (Gensheimer & Narasimhan, 2019; Aalen, 1978). Let $\Delta = 1/J$ be the bin width. For each query sample $i$ and bin $y$, the model outputs a real-valued logit $a_{i,y} \in \mathbb{R}$ and maps it to an increment

$$A_{i,y} = \text{softplus}(a_{i,y}) \cdot \Delta \geq 0. \tag{8}$$

This yields a survival function at the end of bin $y$,

$$S_{i,y} = \exp\Big(-\sum_{m=0}^{y} A_{i,m}\Big). \tag{9}$$

Here $y$ corresponds to the interval $(t_y, t_{y+1}]$, so $S_{i,y}$ is survival at the bin end $t_{y+1}$ (with $S_{i,-1} = 1$ at $t_0 = 0$). The event probability within bin $y$ is $1 - \exp(-A_{i,y})$. Equivalently, the event-time pmf is $p_{i,y} = S_{i,y-1} - S_{i,y}$, and the remaining mass $S_{i,J-1}$ corresponds to $\Pr(T > t_{\text{hor}})$ ("no event within the horizon"). This construction enforces monotonicity by design and yields a proper discrete-time survival distribution with a numerically stable likelihood.

### 3.3. Censoring-consistent likelihood

Let a sample's observed bin be $y$. For an observed event ($\delta = 1$), the negative log-likelihood (NLL) is

$$\mathcal{L}_{\text{event}} = -\log\big(1 - \exp(-A_{i,y})\big) + \sum_{m<y} A_{i,m}. \quad (10)$$

For censoring ($\delta = 0$), we treat the observation as survival through bin $y$:

$$\mathcal{L}_{\text{censor}} = \sum_{m \le y} A_{i,m}. \quad (11)$$

This corresponds to censoring at the bin end $t_{y+1}$. We optimize the mean NLL across query points and tasks. Under conditional independent censoring ($T \perp U \mid x$), this is a proper scoring rule for the discretized conditional event-time distribution; in the prior we also include small fractions of covariate-dependent (still conditionally independent) and weakly dependent censoring for robustness coverage.

### 3.4. PFN-style in-context survival prediction

SurvivalPFN follows the Prior-Data Fitted Network (PFN) framework (Hollmann et al., 2023; Nagler, 2023), approximating the Bayesian posterior predictive distribution induced by a task prior. Let $\tau$ denote a task and $D = D_C \cup D_Q$ a dataset drawn from $p(\cdot \mid \tau)$. Prior-fitting repeatedly samples synthetic tasks $\tau \sim p(\tau)$, draws context and query sets $(D_C, D_Q) \sim p(\cdot \mid \tau)$, and minimizes the expected query objective:

$$\min_{\theta} \mathbb{E}_{\tau \sim p(\tau),\, (D_C, D_Q) \sim p(\cdot|\tau)}$$
$$\left[ \frac{1}{|D_Q|} \sum_{(x,y,\delta) \in D_Q} \ell_\theta(x, y, \delta \mid D_C) \right]. \quad (12)$$

Here $\ell_\theta$ denotes the discrete-time censoring-consistent NLL. The Transformer receives a sequence of context tokens $(x, \tilde{y}, \delta)$ and query tokens with labels masked, preventing leakage and matching inference-time information. We use a context–query attention mask: context tokens attend to each other, and query tokens attend only to context tokens (no query–query attention). At inference time, we use the dataset's training split as context and obtain survival functions for the test split without gradient-based fine-tuning; Fig. 1 illustrates the tokenization and mask.

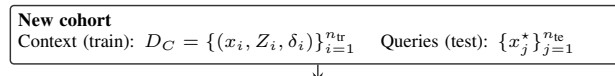

**New cohort**
Context (train): $D_C = \{(x_i, Z_i, \delta_i)\}_{i=1}^{n_{\text{tr}}}$     Queries (test): $\{x_j^\star\}_{j=1}^{n_{\text{te}}}$

**Inference-only in-context prediction**
Sequence: context tokens $(x, Z, \delta)$ and query tokens ($x^\star$, MASK, MASK); context–query mask (queries attend to context; no query–query attention); one forward pass with fixed pretrained weights $f_\theta$ (no per-cohort refit)

**Outputs**
Survival curves $S(t \mid x_j^\star, D_C)$ and risk scores $r(x_j^\star)$ for downstream evaluation

*Figure 1.* SurvivalPFN inference as in-context prediction. The training split forms labeled context $D_C = \{(x_i, Z_i, \delta_i)\}$; test covariates are queries with masked label channels, and a context–query attention mask prevents query–query attention. A single forward pass with fixed pretrained weights yields dataset-conditioned survival curves $S(t \mid x^\star, D_C)$ (and risk scores for ranking) without per-dataset optimization.

### 3.5. Architecture and invariances

SurvivalPFN instantiates the PFN predictor with a Transformer backbone for covariate vectors. Numerical features are standardized using context-only statistics, and categorical features are encoded consistently with baselines. For context points, the discretized outcome ($\tilde{y}, \delta$) is embedded using fixed time-bin indices with learnable bin embeddings and an event-indicator embedding; for query points, we mask the label channels with a dedicated embedding.

### 3.6. Synthetic survival task prior

The pretraining prior $p(\tau)$ targets heterogeneous survival prediction tasks with covariate vectors, with explicit diversity in feature types, effect strengths, and censoring regimes. Figure 2 provides a compact schematic of the synthetic-task generator used for prior-fitting.

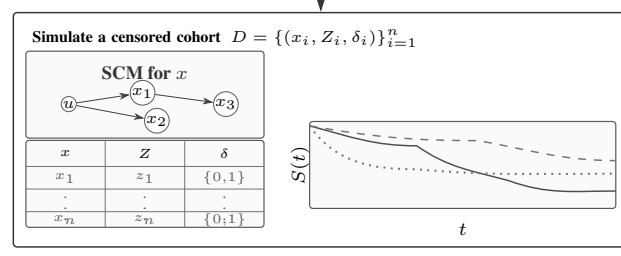

**Synthetic-task prior $p(\tau)$ (mixture)**
Mechanism: PH / non-PH / AFT; baseline hazard: Weibull / random-walk / Fourier.
Covariates: SCM-sampled (cont+cat), sparsity, signal strength; censoring: mostly independent + small dependent variants.

**Simulate a censored cohort** $D = \{(x_i, Z_i, \delta_i)\}_{i=1}^n$

*Figure 2.* Schematic of the synthetic survival-task prior: we sample a mechanism and hyperparameters (PH/non-PH/AFT; hazard families; feature types; censoring regimes), then simulate censored cohorts in the same $(x, Z, \delta)$ format as real data for prior-fitting.

We design $p(\tau)$ to cover diverse regimes: mixed continu-

ous/categorical covariates, varying signal strength and sparsity, event mechanisms spanning PH/non-PH/AFT structure, and censoring rates from near-zero to heavy. Independent right-censoring is the default regime, but we include small fractions of covariate-dependent (still conditionally independent) and weakly dependent censoring for robustness coverage. Appendix A.7 provides the full prior specification and implementation details.

## 4. Experiments

We evaluate SurvivalPFN on the SurvSet benchmark in the small-cohort, heavily censored regime, which is characteristic of data-scarce applications such as medical studies. Our evaluation asks whether a prior-fit, inference-only PFN can (i) match or outperform tuned baselines on discrimination and prediction error, and (ii) produce calibrated survival curves without per-dataset optimization.

### 4.1. Benchmark and protocol

Our primary benchmark uses the SurvSet collection of public survival datasets (Drysdale, 2022), focusing on small-sample time-to-event tasks. The suite spans oncology, clinical, and socio-behavioral domains; Table 1 provides a high-level domain breakdown.

*Table 1.* Application-domain breakdown of the benchmark suite.

| Domain | #Datasets | Share |
|---|---|---|
| Oncology | 24 | 60.0% |
| Clinical | 13 | 32.5% |
| Socio-behavioral | 3 | 7.5% |

Table Table 4 lists the datasets and key statistics, including cohort size and event/censoring fractions. We focus on small cohorts by restricting to SurvSet datasets with at most 1000 instances (no within-dataset subsampling), yielding 40 datasets. We generate three repeated 50/50 train/test splits with event-indicator stratification when feasible. All dataset-dependent preprocessing and evaluation time grids are fit on the training split only and then applied unchanged to the test split. In the main benchmark, the full training split forms the labeled context and the test split serves as queries. We report dataset-equal macro averages over datasets and repeated splits.

### 4.2. Metrics

We report C-index (Harrell et al., 1996), time-dependent AUC and its integrated summary (iAUC) (Heagerty et al., 2000), and the integrated Brier score (IBS) (Graf et al., 1999). We construct $K = 50$ evaluation horizons from training-split quantiles (0.05–0.95) and apply IPCW truncation at a training-selected cutoff time $t_{tr}$ (right-truncating

outcomes at $t_{tr}$) to avoid degenerate IPCW weights, ensuring that $\widehat{G}(t)$ remains positive over the evaluated range (Uno et al., 2011). For discretized models, we evaluate $S(t \mid x)$ via the induced right-continuous step function on bin ends (used for iAUC/IBS). We also report D-calibration (Haider et al., 2020) using the L1 calibration error and the chi-square uniformity test; for interpretability, we summarize the test by its pass rate (fraction of datasets with $p > 0.05$), rather than averaging $p$-values.

### 4.3. Baselines and implementation details

We compare against classical baselines (CoxPH (Cox, 1972), RSF (Ishwaran et al., 2008), GBS (Binder & Schumacher, 2008), and SSVM (Van Belle et al., 2011)). Neural Cox-style baselines include DeepSurv (Katzman et al., 2018) and CoxTime/CoxCC (Kvamme et al., 2019). We also include discrete-time and other neural baselines: DeepHit (Lee et al., 2018), Deep Survival Machines (DSM) (Nagpal et al., 2021), and discretized hazard baselines such as MTLR (Yu et al., 2011), LogisticHazard (Gensheimer & Narasimhan, 2019), PCHazard and PMF (Kvamme & Borgan, 2019). All methods share the same splits, data transformations, and evaluation horizons. Unless stated, we tune baselines via a held-out validation split from the training data (val_frac=0.2), select by validation C-index, and refit on the full training split; for SSVM we tune only its regularization strength. Full hyperparameter grids and training budgets are summarized in the appendix (Appendix A.1). SurvivalPFN uses fixed pretrained weights and predicts the test split without any per-dataset training or tuning. We fix the discretization ($J = 64$) and time-normalization quantile ($q = 0.95$) globally; Appendix Appendix A.2 reports sensitivity to these choices. SurvivalPFN is a Transformer with 13.2 M parameters and uses the latest pretrained checkpoint for all reported results. Code and pretrained weights are available at `https://anonymous.4open.science/r/SurvivalPFN`.

## 5. Results and Discussion

Following Section 4, we report dataset-equal macro averages over 40 SurvSet datasets (three repeated 50/50 splits) in the small-cohort, heavily censored setting, which is dominated by clinical and oncology tasks. We evaluate SurvivalPFN in its intended *inference-only* mode—fixed pretrained weights; one forward pass conditioned on the training split as labeled context—and compare to leakage-free baselines tuned on training-only validation. We report discrimination (C-index/iAUC), prediction error (IBS), calibration, and per-dataset wall-clock.

**Macro performance: strong accuracy without per-cohort optimization.** Table 2 summarizes dataset-equal macro performance across classical and neural baselines.

*Table 2.* Macro-average benchmark results on the evaluation suite (dataset-equal macro). Values are mean±std across datasets.

| Method | C-index↑ | IBS↓ | iAUC↑ |
|---|---|---|---|
| SurvivalPFN | **0.6629±0.0836** | **0.1779±0.0334** | **0.7029±0.0902** |
| CoxPH | 0.6285±0.0846 | 0.2391±0.1040 | 0.6575±0.0925 |
| SSVM | 0.6368±0.0856 | 0.2023±0.0565 | 0.6688±0.0933 |
| RSF | 0.6509±0.0859 | 0.1839±0.0323 | 0.6803±0.0930 |
| GBS | 0.6273±0.0848 | 0.1999±0.0380 | 0.6535±0.0929 |
| DeepSurv | 0.6169±0.0786 | 0.2550±0.0631 | 0.6421±0.0874 |
| CoxTime | 0.6155±0.0784 | 0.2392±0.0532 | 0.6439±0.0835 |
| CoxCC | 0.6229±0.0830 | 0.2355±0.0543 | 0.6483±0.0891 |
| LogisticHazard | 0.6089±0.0648 | 0.4335±0.1647 | 0.6114±0.0689 |
| PCHazard | 0.6101±0.0715 | 0.4217±0.1733 | 0.6120±0.0700 |
| PMF | 0.6030±0.0789 | 0.2371±0.0477 | 0.6096±0.0754 |
| MTLR | 0.6027±0.0733 | 0.2704±0.0620 | 0.6211±0.0783 |
| DeepHit | 0.6055±0.0799 | 0.2348±0.0434 | 0.6087±0.0701 |
| DSM | 0.6287±0.0937 | 0.2150±0.0919 | 0.6611±0.1043 |

SurvivalPFN attains the best averages for discrimination (C-index/iAUC) and prediction error (IBS) while using a single pretrained checkpoint—no per-dataset hyperparameter search, no gradient-based refitting, and no early-stopping decisions on the target cohort. This is the PFN operating point: amortize inductive bias and parts of model selection into one offline prior-fitting run, then reuse a fixed inference recipe across many downstream cohorts.

The advantage is most pronounced over deep-learning baselines that require per-dataset training and validation-driven selection: compared to DeepSurv, SurvivalPFN improves iAUC by $+0.0608$ and reduces IBS by $-0.0771$. This pattern is consistent with the target regime, where event counts are limited and validation-driven model selection is statistically noisy—a setting common in data-scarce, high-stakes applications such as medical studies. Against a strong nonparametric baseline (RSF), SurvivalPFN improves by $+0.0226$ iAUC and $-0.0060$ IBS on average (C-index $+0.0120$); improvements are broadly distributed (27/40 iAUC wins and 29/40 IBS wins; paired sign tests over datasets $p = 0.038$ and $p = 0.006$).

**Compute: inference-only versus per-dataset training.** SurvivalPFN separates one-time pretraining from per-dataset use: once prior-fit, downstream usage is a single forward pass conditioned on the training split as context. Figure 3 visualizes an accuracy–time tradeoff (iAUC/IBS vs. wall-clock) where the x-axis is the per-dataset wall-clock for baselines under a *Train+Predict* definition (fit + predict, excluding hyperparameter tuning), while SurvivalPFN reports inference only. This isolates the cost of *running a method once* on a new cohort; we report tuning time separately in Appendix A.6.

On our suite, SurvivalPFN attains the best macro performance while remaining competitive in per-dataset wall-clock. Inference time is heavy-tailed across datasets (median 0.04 s), with details and the dominant outlier discussed in Appendix A.6. Neural baselines incur per-dataset training costs even when tuned lightly; classical models can be fast

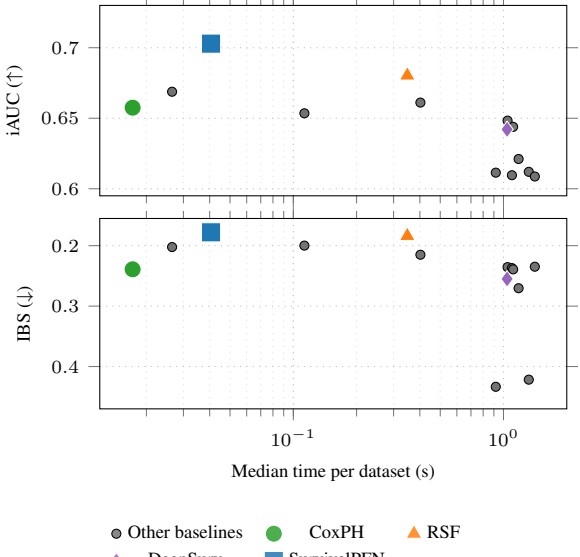

*Figure 3.* Accuracy–time tradeoff on the 40-dataset benchmark: discrimination (iAUC, top; higher is better) and prediction error (IBS, bottom; lower is better) versus median per-dataset wall-clock. Baseline time is Train+Predict (excluding hyperparameter tuning); SurvivalPFN reports inference only with fixed pretrained weights.

but often trail on accuracy or calibration. We separately report the one-time pretraining cost in Appendix A.6; the central claim is that this upfront cost can be amortized across many downstream cohorts to yield a stable inference-only procedure.

**Calibration and curve quality in small cohorts.** Survival analysis requires reliable survival curves, not only risk ranking. Table 3 reports D-calibration summary statistics, including the L1 error and pass rate of the chi-square uniformity test.

*Table 3.* Calibration summary on the 40-dataset benchmark. We report dataset-equal mean±std for D-calib L1 and the fraction of train–test splits with $p>0.05$ in the D-calibration chi-square test (three repeats per dataset).

| Method | D-calib L1 ↓ | D-calib pass@0.05 (splits) ↑ |
|---|---|---|
| SurvivalPFN | **0.187 ± 0.089** | **108/120** |
| RSF | 0.190 ± 0.119 | 106/120 |
| CoxPH | 0.382 ± 0.335 | 59/120 |
| DeepSurv | 0.635 ± 0.332 | 18/117 |
| DeepHit | 0.435 ± 0.192 | 39/120 |

SurvivalPFN attains the lowest calibration error and passes the test on 108/120 train–test splits (three repeats per dataset), slightly improving over RSF (106/120). Notably, representative baselines can fail this calibration check at high rates (e.g., CoxPH 59/120; DeepSurv 18/117, with denominators reflecting successful fits), underscoring that calibration is a primary failure mode in small, censored cohorts. Empirically, flexible baselines can look competitive on dis-

crimination while producing miscalibrated curves, particularly when event counts are low and validation signals are weak. SurvivalPFN mitigates this by (i) enforcing monotone survival through the integrated-hazard parameterization and (ii) training with a censoring-consistent likelihood across diverse synthetic tasks, which acts as an inductive prior for curve shape and uncertainty in the small-data regime. Appendix Figure 6 provides a per-dataset calibration comparison against RSF.

**Transfer: design choices and a context control.** A natural concern for prior-fit survival models is whether synthetic training transfers to heterogeneous real datasets. SurvivalPFN targets transfer via three aligned design choices. (1) *Breadth:* the task prior spans a wide range of censoring fractions, time scales, signal strengths, and cohort sizes (Fig. 2); the full mechanism mix (PH/non-PH/AFT) and sampling hyperparameters are specified in Appendix A.7. (2) *Invariances:* context-derived time normalization and discretization reduce dataset-specific scale and encourage conditioning on relative temporal structure. (3) *Proper supervision:* the integrated-hazard parameterization enforces monotone curves, and the censoring-consistent likelihood provides a proper training signal for the discretized event-time distribution.

Beyond plausibility arguments, we add a simple control aligned with the PFN mechanism. We vary the requested context size at inference and compare (i) true labeled context to (ii) a label-shuffle control that keeps covariates fixed but randomly permutes the labels $(Z, \delta)$ among context points, breaking the covariate–label correspondence. Appendix Figure 7 shows that more correctly labeled context steadily improves iAUC and reduces IBS. Under the label-shuffle control, iAUC shows little scaling with context size, while IBS improves only mildly and quickly plateaus, consistent with estimating marginal survival rather than learning covariate-conditioned risk. This indicates that the main context-scaling gains (especially in discrimination) rely on correctly paired survival labels, rather than benefiting merely from more context tokens/covariates or from test-time leakage.

**Per-dataset heterogeneity: patterns across cohorts.** Macro averages hide substantial heterogeneity across cohorts. Against RSF, SurvivalPFN improves iAUC on 27/40 datasets and IBS on 29/40, and the two improvements typically co-occur (see Appendix Figure 5). The largest discrimination gains appear on several small-to-medium cohorts where per-dataset model selection is intrinsically brittle or where the hazard signal departs from simple proportional effects: for example, `survset_veteran` ($\approx$ +0.18 iAUC), `survset_stagec` ($\approx$ +0.09), and `survset_phpl04k8a` ($\approx$ +0.07). These cases match

our motivation: replacing per-cohort optimization with a stable inference recipe can deliver competitive accuracy without relying on a fragile validation split.

To connect this pattern to the small-cohort motivation more directly, we stratify datasets by cohort size and observe larger average gains in the smallest regime. For $n \leq 200$ (16 datasets), the mean iAUC improvement over RSF increases to +0.0397, compared to +0.0114 for $200 < n \leq 500$ and +0.0106 for $n > 500$. While this analysis is observational, it is consistent with a key PFN hypothesis: when data are scarce, a prior-fit model can supply inductive structure that per-dataset optimization cannot reliably recover.

Conversely, we observe cohorts where RSF remains difficult to beat in discrimination even when IBS improves, and a handful where both iAUC and IBS degrade (e.g., `survset_cgd`, `survset_aml_bull`). Overall, the pattern is consistent with a prior-fit model that is strongest when a coherent synthetic inductive bias can be transferred via context, but can underperform when prior mismatch or extreme feature structure dominates.

**Limitations.** Limitations include extremely small samples or very low event counts, where any method can become unstable. Our pretraining prior and likelihood focus on single-event, right-censored survival settings and do not model competing risks, left truncation, or time-varying covariates; mismatch to these settings can degrade performance. Finally, the Transformer cost grows with context length; while our target regime is small cohorts (here $n \leq 1000$), scaling inference-only PFNs to much larger datasets may require architectural or approximation changes. Extending the prior and likelihood to broader survival settings is a natural direction.

Overall, these results highlight SurvivalPFN as a practical inference-only approach for data-scarce survival modeling: a single pretrained checkpoint can be applied across many small, censored cohorts by conditioning on the labeled training split as context, without per-cohort hyperparameter tuning.

# 6. Conclusion

We introduced SurvivalPFN, a Prior-Data Fitted Network (PFN)-style Transformer for censoring-aware survival prediction that produces survival curves via *in-context inference* rather than per-dataset optimization. SurvivalPFN separates a one-time prior-fitting stage—pretraining on synthetic censored tasks drawn from a broad survival-task prior under a censoring-consistent likelihood—from downstream use, where the training split of a new dataset serves as labeled context and a single forward pass yields dataset-conditioned curves $S(t \mid x, D_C)$.

On 40 SurvSet datasets with $n \leq 1000$, SurvivalPFN achieves the strongest dataset-equal macro performance on discrimination and prediction error (C-index, iAUC, IBS) and the best calibration (D-calibration), despite using fixed pretrained weights with no per-dataset tuning.

Taken together, these results support inference-only PFNs as a practical operating point for data-scarce survival analysis: when cohorts are small and censoring is heavy, validation-driven tuning can be both statistically brittle and operationally costly, and calibration failures become common. This setting is frequent in high-stakes domains such as healthcare, where cohorts can be limited by privacy, rare events, and long follow-up, yet decisions require well-calibrated risk over time.

Future work includes extending the prior and likelihood to competing risks, left truncation, and time-varying covariates; improving scalability beyond the small-cohort regime; and developing domain-specialized priors that encode clinical or reliability structure.

## Impact Statement

Survival prediction can inform high-stakes decisions in medicine, public policy, and reliability engineering. Our work targets a frequent practical regime where cohorts are small and censoring is heavy—settings in which validation-driven model selection can be statistically brittle, operationally costly, and calibration failures become common.

SurvivalPFN aims to reduce this burden by separating a one-time prior-fitting stage from downstream use: once pretrained, it produces dataset-conditioned survival curves via a single forward pass on the cohort's labeled training split as context, without per-dataset hyperparameter tuning or gradient-based refitting.

In our benchmark on 40 SurvSet datasets, this inference-only operating point achieves strong dataset-equal macro performance on discrimination and prediction error and improves calibration relative to representative baselines, suggesting a practical default for data-scarce cohorts where reliable survival curves—not only risk ranking—are required.

Potential negative impacts include misuse of predicted survival curves as automated decision rules, inappropriate deployment under dataset shift, and amplification of historical bias when covariates encode inequities.

Because SurvivalPFN is prior-fit on synthetic tasks, prior mismatch can manifest as miscalibrated curves or misleading uncertainty estimates; additionally, our prior and likelihood focus on single-event, right-censored settings and do not model competing risks, left truncation, or time-varying covariates. We therefore recommend using SurvivalPFN strictly as a decision-support component with domain exper-tise and human oversight: perform dataset-specific validation (including calibration checks), monitor for drift, audit covariates for leakage and bias, and clearly communicate uncertainty and known limitations to stakeholders.

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

*Table 4.* Benchmark datasets, coarse application domains, and key statistics. Event/censoring fractions are computed on the full dataset. $d$ is post-encoding dimensionality (after one-hot), while the raw feature cap is applied before encoding. $t_{tr}$ denotes the IPCW truncation time (min–max across splits).

| Dataset | Domain | $n$ | $d$ | Event% | Censor% | $t_{tr}$ (min–max) |
|---|---|---|---|---|---|---|
| survset_grace | Clinical | 1000 | 11 | 32.4% | 67.6% | 180–180 |
| survset_colon | Oncology | 929 | 58 | 48.7% | 51.3% | 3309–3325 |
| survset_dataovarian1 | Oncology | 912 | 73 | 59.6% | 40.4% | 3644–4980 |
| survset_gbsg2 | Oncology | 686 | 46 | 43.6% | 56.4% | 2556–2612 |
| survset_uis | Socio-behavioral | 628 | 88 | 80.9% | 19.1% | 762–805 |
| survset_vlbw | Clinical | 617 | 142 | 17.3% | 82.7% | 442–797 |
| survset_follic | Oncology | 541 | 11 | 64.3% | 35.7% | 28.8–29.3 |
| survset_cost | Clinical | 518 | 43 | 78.0% | 22.0% | 4259–4259 |
| survset_prostate | Oncology | 502 | 88 | 70.5% | 29.5% | 75.0–75.0 |
| survset_whas500 | Clinical | 461 | 67 | 38.2% | 61.8% | 2178–2353 |
| survset_unemployment | Socio-behavioral | 452 | 24 | 56.6% | 43.4% | 112–113 |
| survset_php104k8a | Clinical | 442 | 5 | 53.4% | 46.6% | 13.5–14.7 |
| survset_zinc | Clinical | 431 | 90 | 18.8% | 81.2% | 5980–5980 |
| survset_diabetes | Clinical | 394 | 13 | 39.3% | 60.7% | 75.0–75.0 |
| survset_retinopathy | Clinical | 394 | 35 | 39.3% | 60.7% | 74.9–74.9 |
| survset_ova | Oncology | 358 | 34 | 74.3% | 25.7% | 2596–2661 |
| survset_pbc3 | Clinical | 349 | 41 | 17.5% | 82.5% | 2118–2131 |
| survset_pbc | Clinical | 312 | 21 | 40.1% | 59.9% | 4459–4523 |
| survset_dbcd | Oncology | 295 | 22 | 26.8% | 73.2% | 17.6–18.1 |
| survset_e1684 | Oncology | 284 | 9 | 69.0% | 31.0% | 9.38–9.38 |
| survset_mgus | Oncology | 241 | 58 | 93.4% | 6.6% | 13019–14111 |
| survset_cancer | Oncology | 228 | 67 | 72.4% | 27.6% | 840–965 |
| survset_hepatocellular | Oncology | 227 | 81 | 42.7% | 57.3% | 76.0–81.0 |
| survset_melanoma | Oncology | 205 | 21 | 27.8% | 72.2% | 4668–4926 |
| survset_wpbc | Oncology | 198 | 50 | 23.7% | 76.3% | 119–123 |
| survset_d_oropha_rec | Oncology | 192 | 69 | 72.4% | 27.6% | 4.31–4.84 |
| survset_burn | Clinical | 154 | 74 | 31.2% | 68.8% | 62.0–71.0 |
| survset_stagec | Oncology | 146 | 71 | 37.0% | 63.0% | 15.9–16.7 |
| survset_nki70 | Oncology | 144 | 38 | 33.3% | 66.7% | 17.4–17.4 |
| survset_veteran | Oncology | 137 | 80 | 93.4% | 6.6% | 991–991 |
| survset_cgd | Clinical | 128 | 87 | 34.4% | 65.6% | 364–388 |
| survset_pharmacosmoking | Socio-behavioral | 125 | 114 | 71.2% | 28.8% | 182–182 |
| survset_micro_censure | Oncology | 117 | 239 | 22.2% | 77.8% | 6.13–6.45 |
| survset_aml_bull | Oncology | 116 | 1675 | 57.8% | 42.2% | 1336–1388 |
| survset_nsbcd | Oncology | 115 | 540 | 33.0% | 67.0% | 92.0–95.0 |
| survset_breast | Oncology | 100 | 16 | 26.0% | 74.0% | 72.0–72.0 |
| survset_z243 | Clinical | 100 | 60 | 100.0% | 0.0% | 28.0–32.0 |
| survset_mclcleaned | Oncology | 92 | 375 | 69.6% | 30.4% | 9.23–13.8 |
| survset_bergamaschi | Oncology | 82 | 8 | 34.1% | 65.9% | 6.67–7.42 |
| survset_glioma | Oncology | 37 | 30 | 62.2% | 37.8% | 58.0–61.0 |

# A. Additional Details

## A.1. Reproducibility and protocol summary

We follow an evaluation protocol that derives data-transformation and evaluation choices from the training split only and summarize the key choices here to facilitate reproducibility checks. For each dataset we generate three repeated 50/50 train–test splits with event-indicator stratification when feasible. We restrict the benchmark to datasets with at most 1000 instances (no within-dataset subsampling). All transformations (numerical standardization and categorical encoding) are fit on the training split only and then applied to the test split.

Our pretraining objective is the censoring-consistent discrete-time hazard negative log-likelihood (NLL) under a fixed discretization. For time-dependent evaluation we use $K = 50$ horizons selected from training-split quantiles (0.05–0.95). For IPCW-based metrics we estimate the censoring distribution on the training split only and right-truncate test outcomes at a training-selected time $t_{tr}$ (treating $T > t_{tr}$ as censored at $t_{tr}$), ensuring that IPCW estimators remain well-defined without accessing test outcomes. SurvivalPFN performs amortized downstream inference without gradient-based fine-tuning: it conditions on the training split as labeled context. Pretraining runs for 1000 epochs with 200 steps per epoch (200k updates) using mixed precision. We use AdamW with constant learning rate ($9 \times 10^{-5}$), weight decay 0.035, no warmup, and gradient clipping at 1.0; the nominal batch size is 64 with dynamic packing subject to the cell/sequence budgets. Baselines are tuned per dataset via a held-out validation split from the training data (val_frac=0.2), followed by refitting the selected configuration on the full training split. For neural baselines, each learning-rate candidate is trained for a fixed 200-epoch budget on the tuning split; the selected learning rate is then refit with the full training budget listed below. The selection metric is validation C-index, except for SSVM where we select the regularization strength via grid search using training-set C-index. When a split is too small or has insufficient events to form a valid validation set, tuning is skipped and the method falls back to default settings. Default model structures are as follows. MTLR, LogisticHazard, PCHazard, PMF, DeepHit, DeepSurv, CoxTime, and CoxCC use MLPs with hidden sizes (128, 64), batch normalization, dropout $d_0 = 0.1$, and $lr_0 = 10^{-3}$,

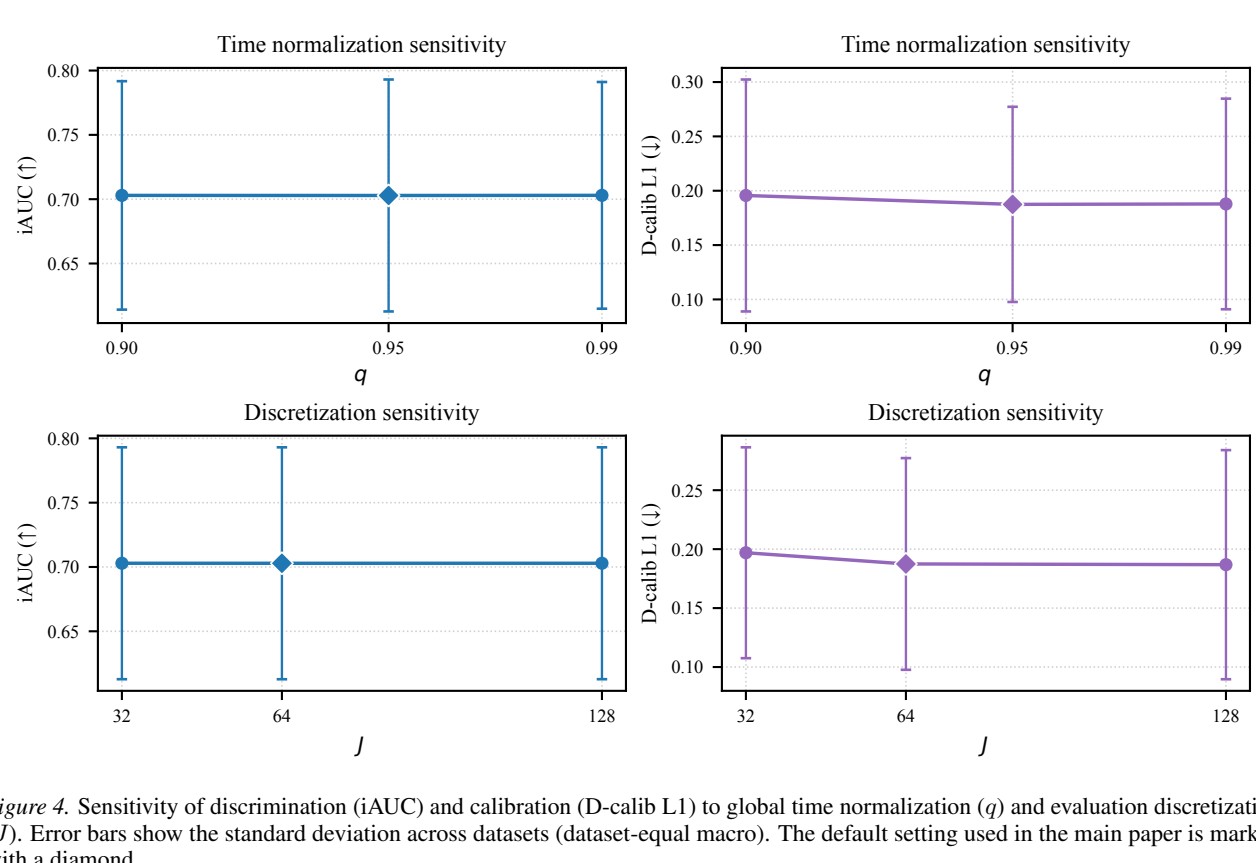

*Figure 4.* Sensitivity of discrimination (iAUC) and calibration (D-calib L1) to global time normalization ($q$) and evaluation discretization ($J$). Error bars show the standard deviation across datasets (dataset-equal macro). The default setting used in the main paper is marked with a diamond.

*Table 5.* Sensitivity to the discretization resolution $J$ (dataset-equal macro). We re-bin the predicted piecewise-constant hazard representation for evaluation. The default setting used elsewhere is marked with $\star$.

| Discretization | C-index↑ | IBS↓ | iAUC↑ |
|---|---|---|---|
| $J = 32$ | **0.6629±0.0836** | 0.1782±0.0331 | **0.7029±0.0902** |
| $J = 64^{\star}$ | **0.6629±0.0836** | **0.1779±0.0334** | **0.7029±0.0902** |
| $J = 128$ | 0.6628±0.0836 | **0.1779±0.0335** | **0.7029±0.0901** |

trained with Adam for 1000 epochs with batch size 256. DeepHit uses $\alpha = 0.2$ and $\sigma = 0.1$. SSVM uses $\alpha_0 = 1.0$ and 20 optimizer iterations. DSM uses $k_0 = 3$ mixture components, layers (100), Weibull components, temperature 1.0, $lr_0 = 10^{-3}$, and 50 training iterations. Hyperparameter grids are lightweight and use only training data. For neural baselines, we grid-search $lr \in \{0.1, 0.3, 1.0, 3.0\} \cdot lr_0$ using a fixed 200-epoch tuning budget per candidate. For CoxPH we grid-search $\alpha \in \{0.1, 0.3, 1, 3, 10, 30, 100\} \cdot \alpha_0$. For SSVM we grid-search $\alpha \in \{0.01, 0.03, 0.1, 0.3, 1, 3, 10, 30, 100\} \cdot \alpha_0$. For GBS we grid-search $\eta \in \{0.1, 0.3, 1.0, 3.0\} \cdot \eta_0$ and $n_{estimators} \in \{0.5, 1, 2, 4\} \cdot n_0$. For RSF we use $n_0 = 1000$, $min\_leaf_0 = 15$, and $min\_split_0 = 10$.

## A.2. Sensitivity to global time normalization and discretization

SurvivalPFN is intended to run with a fixed, global configuration at deployment. We therefore check the sensitivity of performance and calibration to two global choices: the time-normalization quantile $q$ (used to map per-dataset time scales to a shared range) and the evaluation discretization resolution $J$ (used to evaluate survival curves from piecewise-constant hazards). Figure 4 plots dataset-equal macro iAUC and D-calibration L1 for $q \in \{0.90, 0.95, 0.99\}$ and $J \in \{32, 64, 128\}$, using the same SurvSet-40 benchmark and three repeated splits as in the main results. We also include the corresponding summary tables (Tables 5 and 6); across the tested range, both discrimination and calibration vary only mildly, supporting our "no knobs at deployment" setting.

*Table 6.* Sensitivity to the time-normalization quantile $q$ (dataset-equal macro). The default setting used elsewhere is marked with $\star$.

| Time scale | C-index↑ | IBS↓ | iAUC↑ |
|---|---|---|---|
| $q = 0.90$ | 0.6631±0.0834 | 0.1780±0.0334 | 0.7029±0.0888 |
| $q = 0.95^{\star}$ | 0.6629±0.0836 | 0.1779±0.0334 | 0.7029±0.0902 |
| $q = 0.99$ | **0.6633±0.0832** | **0.1777±0.0330** | **0.7030±0.0881** |

## A.3. Per-dataset comparison to RSF

To complement macro averages, Figure 5 shows per-dataset scatter plots of SurvivalPFN vs. RSF (mean over splits) for both iAUC and IBS. We observe 27/40 iAUC wins and 29/40 IBS wins. Across datasets, the two deltas are strongly correlated (Pearson correlation $\approx -0.78$ for $\Delta$ iAUC vs. $\Delta$ IBS), and 24/40 datasets improve both metrics simultaneously.

## A.4. Per-dataset calibration comparison

To make calibration differences more concrete, Figure 6 compares per-dataset D-calibration L1 errors (mean over splits) between SurvivalPFN and RSF. Points below the diagonal indicate lower calibration error for SurvivalPFN.

## A.5. Full per-dataset benchmark results

For completeness, Table 7 reports the full per-dataset results for all methods and metrics on the SurvSet benchmark (three repeated 50/50 splits). The table is generated from the same leakage-free protocol as the main results and is intended to support detailed cohort-level inspection.

*Table 7.* Per-dataset benchmark results on real-world survival datasets ($n \leq 1000$), using repeated 50/50 train-test splits. Evaluation horizons for IBS/iAUC are defined from training-set quantiles. Numbers are mean±std across split seeds; best method per row is bolded.

| Dataset | $n$ Metric | PFN | CoxPH | CoxTime | DSM | RSF | GBS | DeepSurv | DeepHit |
|---|---|---|---|---|---|---|---|---|---|
| survset_grace | 1000 C-index ↑ | **.721±.024** | .708±.004 | .703±.011 | .713±.017 | .705±.017 | .686±.030 | .681±.019 | .672±.051 |
| survset_grace | 1000 IBS ↓ | **.167±.008** | .170±.003 | .174±.006 | .177±.004 | .172±.006 | .182±.008 | .197±.023 | .271±.030 |
| survset_grace | 1000 iAUC ↑ | **.813±.013** | .749±.015 | .765±.024 | .782±.015 | .800±.015 | .725±.018 | .727±.030 | .703±.013 |
| survset_colon | 929 C-index ↑ | .643±.014 | .609±.012 | .592±.019 | .639±.007 | **.647±.005** | .609±.029 | .590±.027 | .582±.020 |
| survset_colon | 929 IBS ↓ | .202±.007 | .214±.008 | .260±.026 | .204±.004 | **.199±.008** | .221±.021 | .273±.007 | .248±.027 |
| survset_colon | 929 iAUC ↑ | .697±.020 | .653±.029 | .615±.016 | **.701±.006** | .698±.020 | .644±.054 | .623±.040 | .594±.025 |
| survset_dataovarian1 | 912 C-index ↑ | **.644±.005** | .611±.015 | .594±.007 | .545±.077 | .637±.005 | .599±.022 | .603±.020 | .589±.013 |
| survset_dataovarian1 | 912 IBS ↓ | **.169±.003** | .189±.012 | .205±.002 | .444±.237 | .172±.001 | .199±.028 | .235±.004 | .215±.010 |
| survset_dataovarian1 | 912 iAUC ↑ | **.697±.005** | .649±.018 | .628±.005 | .562±.108 | .684±.009 | .634±.032 | .637±.026 | .610±.017 |
| survset_gbsg2 | 686 C-index ↑ | **.667±.005** | .665±.008 | .590±.025 | .642±.026 | .661±.006 | .660±.029 | .603±.016 | .588±.021 |
| survset_gbsg2 | 686 IBS ↓ | **.188±.007** | .192±.004 | .267±.044 | .195±.007 | .194±.006 | .208±.025 | .266±.014 | .243±.019 |
| survset_gbsg2 | 686 iAUC ↑ | **.719±.012** | .712±.014 | .630±.038 | .695±.029 | .703±.020 | .707±.023 | .629±.021 | .615±.022 |
| survset_uis | 628 C-index ↑ | **.584±.020** | .569±.024 | .553±.001 | .574±.009 | .574±.005 | .571±.019 | .554±.009 | .535±.009 |
| survset_uis | 628 IBS ↓ | **.187±.003** | .215±.019 | .273±.044 | .189±.005 | .187±.004 | .205±.018 | .280±.008 | .254±.054 |
| survset_uis | 628 iAUC ↑ | **.622±.034** | .604±.031 | .581±.011 | .614±.018 | .601±.006 | .612±.030 | .584±.011 | .550±.013 |
| survset_vlbw | 617 C-index ↑ | .908±.015 | .862±.035 | .901±.023 | **.912±.018** | .873±.029 | .901±.031 | .843±.058 | .859±.027 |
| survset_vlbw | 617 IBS ↓ | **.095±.007** | .124±.004 | .135±.029 | .112±.007 | .112±.008 | .113±.016 | .136±.006 | .126±.009 |
| survset_vlbw | 617 iAUC ↑ | .900±.019 | .853±.028 | .889±.036 | **.905±.009** | .862±.028 | .875±.031 | .831±.014 | .690±.071 |
| survset_follic | 541 C-index ↑ | .603±.014 | **.615±.012** | .566±.034 | .605±.013 | .589±.026 | .576±.033 | .565±.012 | .559±.021 |
| survset_follic | 541 IBS ↓ | .199±.007 | **.195±.008** | .233±.034 | .196±.009 | .201±.010 | .225±.045 | .218±.009 | .215±.012 |
| survset_follic | 541 iAUC ↑ | .645±.027 | .658±.026 | .572±.039 | **.664±.005** | .641±.034 | .611±.043 | .589±.021 | .552±.023 |
| survset_cost | 518 C-index ↑ | .678±.008 | .673±.022 | .637±.014 | **.681±.013** | .669±.013 | .638±.017 | – | .635±.016 |
| survset_cost | 518 IBS ↓ | .172±.003 | **.171±.012** | .210±.018 | .173±.005 | .180±.002 | .217±.025 | – | .194±.006 |
| survset_cost | 518 iAUC ↑ | .739±.012 | .729±.026 | .683±.020 | **.747±.021** | .731±.010 | .692±.017 | – | .639±.050 |
| survset_prostate | 502 C-index ↑ | .620±.014 | .578±.010 | .581±.025 | .500±.000 | **.640±.027** | .594±.024 | .578±.010 | .572±.026 |
| survset_prostate | 502 IBS ↓ | .192±.003 | .222±.009 | .289±.003 | .470±.019 | **.190±.002** | .206±.021 | .268±.014 | .251±.026 |
| survset_prostate | 502 iAUC ↑ | .657±.016 | .606±.009 | .605±.023 | .500±.000 | **.670±.024** | .625±.034 | .613±.020 | .588±.035 |
| survset_whas500 | 461 C-index ↑ | .750±.004 | .727±.007 | .691±.030 | .748±.011 | **.762±.008** | .748±.012 | .685±.007 | .691±.017 |
| survset_whas500 | 461 IBS ↓ | **.158±.003** | .178±.004 | .222±.021 | .173±.012 | .162±.003 | .177±.016 | .227±.004 | .182±.024 |
| survset_whas500 | 461 iAUC ↑ | **.786±.013** | .765±.015 | .720±.038 | .782±.018 | .785±.012 | .779±.022 | .725±.018 | .723±.030 |
| survset_unemployment | 452 C-index ↑ | .533±.022 | **.541±.011** | .521±.010 | .506±.009 | .529±.026 | .521±.036 | .507±.038 | .507±.031 |
| survset_unemployment | 452 IBS ↓ | **.210±.014** | .210±.015 | .263±.023 | .326±.211 | .211±.015 | .210±.014 | .255±.021 | .253±.009 |
| survset_unemployment | 452 iAUC ↑ | .552±.037 | **.560±.012** | .520±.012 | .511±.034 | .554±.028 | .521±.046 | .494±.042 | .522±.017 |
| survset_php104k8a | 442 C-index ↑ | .570±.018 | **.575±.019** | .521±.005 | .548±.044 | .534±.014 | .533±.024 | .516±.006 | .511±.024 |
| survset_php104k8a | 442 IBS ↓ | **.204±.014** | .204±.012 | .224±.014 | .207±.013 | .217±.015 | .232±.047 | .221±.019 | .217±.014 |

| Dataset | $n$ Metric | PFN | CoxPH | CoxTime | DSM | RSF | GBS | DeepSurv | DeepHit |
|---|---|---|---|---|---|---|---|---|---|
| survset_php104k8a | 442iAUC ↑ | .599±.038 | **.606±.039** | .523±.011 | .568±.068 | .529±.015 | .537±.027 | .509±.024 | .493±.030 |
| survset_zinc | 431C-index ↑ | **.788±.037** | .727±.107 | .712±.022 | .724±.036 | .780±.037 | .765±.043 | .648±.087 | .683±.004 |
| survset_zinc | 431IBS ↓ | .105±.004 | .128±.036 | .145±.028 | .118±.003 | **.105±.003** | .119±.017 | .165±.041 | .182±.043 |
| survset_zinc | 431iAUC ↑ | **.815±.027** | .766±.095 | .720±.022 | .749±.045 | .781±.037 | .800±.026 | .657±.116 | .599±.029 |
| survset_diabetes | 394C-index ↑ | .590±.018 | .578±.040 | .554±.042 | .538±.049 | **.602±.025** | .599±.029 | .554±.004 | .564±.058 |
| survset_diabetes | 394IBS ↓ | .197±.006 | .199±.004 | .230±.015 | .204±.009 | **.195±.001** | .241±.034 | .244±.018 | .236±.025 |
| survset_diabetes | 394iAUC ↑ | .597±.021 | .583±.036 | .552±.054 | .539±.044 | .596±.020 | **.610±.037** | .538±.030 | .539±.041 |
| survset_retinopathy | 394C-index ↑ | .638±.024 | .637±.017 | .592±.057 | .631±.007 | **.650±.018** | .603±.031 | .539±.027 | .573±.081 |
| survset_retinopathy | 394IBS ↓ | .188±.006 | .190±.002 | .243±.040 | .196±.009 | **.187±.005** | .242±.036 | .294±.033 | .236±.006 |
| survset_retinopathy | 394iAUC ↑ | .673±.008 | .671±.014 | .615±.055 | .661±.020 | **.673±.013** | .623±.040 | .533±.040 | .566±.072 |
| survset_ova | 358C-index ↑ | **.651±.019** | .641±.010 | .611±.014 | .625±.014 | .647±.011 | .642±.013 | .610±.019 | .564±.020 |
| survset_ova | 358IBS ↓ | .186±.014 | **.184±.018** | .203±.019 | .195±.013 | .189±.013 | .191±.009 | .205±.012 | .221±.026 |
| survset_ova | 358iAUC ↑ | **.712±.021** | .695±.013 | .686±.048 | .687±.042 | .700±.005 | .694±.015 | .644±.035 | .595±.055 |
| survset_pbc3 | 349C-index ↑ | **.828±.013** | .790±.037 | .758±.013 | .787±.042 | .797±.021 | .788±.027 | .785±.016 | .750±.049 |
| survset_pbc3 | 349IBS ↓ | .101±.004 | .108±.010 | .128±.015 | .113±.008 | .107±.005 | **.101±.002** | .132±.015 | .122±.014 |
| survset_pbc3 | 349iAUC ↑ | **.861±.006** | .814±.021 | .810±.016 | .816±.045 | .816±.024 | .815±.032 | .812±.019 | .764±.037 |
| survset_pbc | 312C-index ↑ | **.762±.039** | .757±.049 | .746±.013 | .751±.036 | .710±.031 | .734±.027 | .719±.013 | .719±.032 |
| survset_pbc | 312IBS ↓ | .163±.013 | **.162±.023** | .172±.016 | .172±.004 | .175±.013 | .178±.003 | .192±.009 | .197±.019 |
| survset_pbc | 312iAUC ↑ | **.797±.041** | .795±.058 | .748±.023 | .778±.029 | .778±.027 | .787±.041 | .739±.030 | .682±.037 |
| survset_dbcd | 295C-index ↑ | **.647±.056** | .589±.047 | .618±.037 | .550±.124 | .638±.014 | .569±.075 | .618±.037 | .624±.083 |
| survset_dbcd | 295IBS ↓ | .174±.012 | .201±.033 | .213±.014 | .184±.011 | **.170±.008** | .189±.012 | .267±.063 | .209±.050 |
| survset_dbcd | 295iAUC ↑ | .679±.062 | .595±.069 | .663±.040 | .562±.126 | **.681±.036** | .583±.089 | .637±.022 | .676±.044 |
| survset_e1684 | 284C-index ↑ | .529±.012 | .524±.008 | .513±.029 | .515±.034 | .537±.005 | **.553±.034** | .514±.022 | .491±.034 |
| survset_e1684 | 284IBS ↓ | **.220±.003** | .225±.002 | .259±.033 | .224±.030 | .227±.013 | .252±.017 | .255±.005 | .255±.005 |
| survset_e1684 | 284iAUC ↑ | .551±.021 | .541±.020 | .533±.023 | .548±.018 | .545±.006 | **.586±.050** | .536±.038 | .509±.025 |
| survset_mgus | 241C-index ↑ | **.690±.015** | .666±.028 | .662±.017 | .668±.022 | .680±.021 | .676±.019 | .666±.031 | .643±.032 |
| survset_mgus | 241IBS ↓ | .146±.006 | .158±.019 | .179±.005 | .159±.005 | **.145±.011** | .162±.039 | .187±.010 | .184±.012 |
| survset_mgus | 241iAUC ↑ | **.772±.029** | .739±.038 | .741±.019 | .772±.028 | .751±.028 | .754±.038 | .734±.027 | .680±.040 |
| survset_cancer | 228C-index ↑ | .618±.011 | .587±.018 | .580±.046 | **.648±.023** | .641±.033 | .582±.026 | .568±.016 | .555±.008 |
| survset_cancer | 228IBS ↓ | .208±.023 | .262±.057 | .285±.049 | **.207±.028** | .212±.034 | .232±.029 | .320±.053 | .278±.052 |
| survset_cancer | 228iAUC ↑ | .640±.009 | .608±.022 | .600±.030 | .671±.021 | **.674±.021** | .594±.043 | .580±.027 | .553±.028 |
| survset_hepatocellular | 227C-index ↑ | .679±.035 | .600±.047 | .616±.040 | **.695±.027** | .676±.032 | .647±.028 | .611±.045 | .607±.023 |
| survset_hepatocellular | 227IBS ↓ | **.188±.016** | .270±.034 | .288±.024 | .194±.009 | .195±.014 | .199±.014 | .292±.033 | .255±.024 |
| survset_hepatocellular | 227iAUC ↑ | .752±.036 | .630±.067 | .690±.037 | **.780±.022** | .738±.031 | .699±.030 | .673±.040 | .665±.010 |
| survset_melanoma | 205C-index ↑ | .712±.023 | .676±.006 | .663±.052 | .690±.057 | **.739±.019** | .657±.015 | .650±.073 | .645±.037 |
| survset_melanoma | 205IBS ↓ | .155±.012 | .185±.002 | .209±.025 | .251±.174 | **.151±.007** | .210±.080 | .246±.046 | .234±.075 |
| survset_melanoma | 205iAUC ↑ | .756±.032 | .699±.007 | .712±.052 | .725±.079 | **.787±.025** | .691±.022 | .686±.073 | .663±.045 |
| survset_wpbc | 198C-index ↑ | **.690±.021** | .522±.017 | .613±.042 | .682±.031 | .670±.018 | .596±.023 | .641±.046 | .614±.013 |
| survset_wpbc | 198IBS ↓ | **.171±.011** | .309±.059 | .219±.036 | .171±.014 | .173±.012 | .188±.009 | .278±.048 | .239±.030 |
| survset_wpbc | 198iAUC ↑ | **.737±.048** | .512±.031 | .646±.040 | .736±.041 | .697±.042 | .625±.036 | .680±.058 | .625±.040 |
| survset_d_oropha_rec | 192C-index ↑ | **.672±.027** | .623±.042 | .587±.036 | .599±.035 | .651±.038 | .658±.012 | .603±.033 | .575±.031 |
| survset_d_oropha_rec | 192IBS ↓ | **.212±.013** | .240±.019 | .294±.018 | .234±.023 | .224±.017 | .223±.024 | .302±.014 | .292±.028 |
| survset_d_oropha_rec | 192iAUC ↑ | **.754±.032** | .691±.064 | .637±.039 | .661±.053 | .714±.028 | .724±.019 | .651±.051 | .630±.035 |
| survset_burn | 154C-index ↑ | .553±.082 | .554±.049 | .553±.070 | .541±.029 | .503±.025 | **.572±.025** | .547±.042 | .483±.044 |
| survset_burn | 154IBS ↓ | **.201±.036** | .292±.056 | .309±.054 | .202±.029 | .201±.027 | .217±.008 | .334±.074 | .333±.059 |
| survset_burn | 154iAUC ↑ | **.586±.114** | .549±.051 | .566±.034 | .538±.032 | .525±.036 | .547±.023 | .582±.075 | .474±.062 |
| survset_stagec | 146C-index ↑ | **.712±.022** | .568±.024 | .608±.029 | .621±.117 | .660±.025 | .668±.031 | .653±.020 | .636±.027 |
| survset_stagec | 146IBS ↓ | **.193±.024** | .353±.067 | .287±.046 | .199±.023 | .197±.020 | .232±.017 | .312±.015 | .287±.080 |
| survset_stagec | 146iAUC ↑ | **.735±.020** | .587±.044 | .661±.032 | .638±.135 | .643±.033 | .679±.038 | .687±.022 | .654±.006 |
| survset_nki70 | 144C-index ↑ | .654±.038 | .598±.066 | .579±.042 | .624±.092 | **.709±.046** | .610±.055 | .638±.025 | .599±.010 |
| survset_nki70 | 144IBS ↓ | **.192±.032** | .283±.029 | .289±.045 | .196±.030 | .198±.026 | .217±.035 | .269±.049 | .252±.063 |
| survset_nki70 | 144iAUC ↑ | .646±.064 | .613±.067 | .658±.068 | .649±.082 | .621±.079 | .622±.073 | **.664±.049** | .655±.016 |
| survset_veteran | 137C-index ↑ | **.671±.028** | .576±.038 | .633±.030 | .588±.024 | .558±.032 | .626±.037 | .648±.042 | .569±.035 |
| survset_veteran | 137IBS ↓ | **.160±.018** | .316±.044 | .209±.021 | .163±.016 | .173±.024 | .170±.033 | .189±.014 | .243±.024 |
| survset_veteran | 137iAUC ↑ | **.740±.034** | .594±.053 | .673±.032 | .653±.102 | .563±.021 | .666±.044 | .692±.063 | .594±.045 |
| survset_cgd | 128C-index ↑ | .565±.015 | .533±.039 | .510±.052 | .566±.074 | **.603±.019** | .511±.051 | .522±.024 | .513±.037 |
| survset_cgd | 128IBS ↓ | .172±.015 | .278±.009 | .265±.082 | .171±.014 | **.163±.009** | .188±.015 | .290±.066 | .252±.033 |
| survset_cgd | 128iAUC ↑ | .589±.046 | .552±.087 | .509±.074 | .572±.084 | **.660±.023** | .525±.088 | .523±.032 | .511±.037 |
| survset_pharmacosmoking | 125C-index ↑ | .562±.007 | .543±.056 | .541±.054 | .500±.000 | **.597±.017** | .568±.017 | .524±.059 | .513±.017 |
| survset_pharmacosmoking | 125IBS ↓ | .228±.011 | .549±.022 | .353±.067 | .559±.024 | **.224±.011** | .246±.023 | .366±.036 | .292±.054 |
| survset_pharmacosmoking | 125iAUC ↑ | .572±.011 | .566±.090 | .555±.078 | .500±.000 | **.623±.032** | .586±.035 | .527±.108 | .536±.004 |
| survset_micro_censure | 117C-index ↑ | .669±.062 | .528±.216 | .581±.136 | .631±.115 | **.680±.083** | .596±.043 | .618±.195 | .585±.169 |
| survset_micro_censure | 117IBS ↓ | .176±.019 | .235±.034 | .270±.094 | .177±.011 | **.173±.010** | .188±.021 | .283±.044 | .221±.078 |
| survset_micro_censure | 117iAUC ↑ | **.702±.090** | .524±.107 | .618±.156 | .658±.098 | .688±.098 | .586±.031 | .662±.202 | .632±.186 |
| survset_aml_bull | 116C-index ↑ | .599±.102 | .517±.065 | .577±.063 | .552±.073 | **.626±.050** | .485±.063 | .568±.054 | .601±.073 |
| survset_aml_bull | 116IBS ↓ | .195±.036 | .512±.017 | .270±.080 | .300±.169 | **.188±.035** | .241±.033 | .281±.074 | .215±.049 |

| Dataset | $n$ Metric | PFN | CoxPH | CoxTime | DSM | RSF | GBS | DeepSurv | DeepHit |
|---|---|---|---|---|---|---|---|---|---|
| survset_aml_bull | 116iAUC ↑ | .647±.095 | .535±.073 | .589±.112 | .543±.040 | **.706±.027** | .474±.098 | .611±.067 | .627±.082 |
| survset_nsbcd | 115C-index ↑ | .728±.022 | .714±.049 | .693±.027 | .727±.015 | **.747±.010** | .619±.140 | .731±.060 | .695±.113 |
| survset_nsbcd | 115IBS ↓ | **.185±.034** | .348±.014 | .275±.110 | .188±.032 | .187±.036 | .226±.066 | .400±.187 | .233±.063 |
| survset_nsbcd | 115iAUC ↑ | .757±.033 | .760±.040 | .725±.051 | .732±.022 | **.784±.023** | .634±.178 | .755±.075 | .657±.068 |
| survset_breast | 100C-index ↑ | .800±.041 | .789±.053 | .669±.040 | .716±.101 | **.808±.044** | .722±.037 | .689±.060 | .738±.078 |
| survset_breast | 100IBS ↓ | **.108±.008** | .110±.013 | .143±.017 | .116±.008 | .125±.008 | .119±.008 | .137±.019 | .191±.068 |
| survset_breast | 100iAUC ↑ | .813±.040 | .797±.061 | .679±.040 | .715±.138 | **.825±.047** | .711±.060 | .671±.091 | .672±.035 |
| survset_z243 | 100C-index ↑ | .557±.015 | **.564±.004** | .511±.048 | .536±.029 | .529±.050 | .494±.043 | .498±.075 | .515±.031 |
| survset_z243 | 100IBS ↓ | **.148±.002** | .187±.016 | .236±.023 | .151±.003 | .149±.002 | .171±.014 | .252±.017 | .280±.006 |
| survset_z243 | 100iAUC ↑ | .572±.010 | **.607±.011** | .524±.071 | .568±.051 | .535±.017 | .508±.054 | .504±.096 | .505±.032 |
| survset_mclcleaned | 92C-index ↑ | .645±.081 | .628±.075 | .632±.069 | .561±.022 | **.688±.046** | .561±.030 | .619±.056 | .559±.064 |
| survset_mclcleaned | 92IBS ↓ | **.211±.030** | .535±.070 | .302±.074 | .233±.042 | .227±.030 | .245±.040 | .310±.058 | .259±.032 |
| survset_mclcleaned | 92iAUC ↑ | .705±.091 | .655±.102 | .664±.085 | .628±.045 | **.762±.050** | .581±.070 | .637±.107 | .608±.079 |
| survset_bergamaschi | 82C-index ↑ | **.642±.078** | .634±.051 | .582±.109 | .506±.069 | .589±.018 | .633±.031 | .610±.074 | .606±.114 |
| survset_bergamaschi | 82IBS ↓ | .239±.020 | **.219±.025** | .315±.097 | .259±.007 | .251±.012 | .260±.025 | .374±.068 | .307±.022 |
| survset_bergamaschi | 82iAUC ↑ | .666±.110 | .648±.076 | .602±.080 | .503±.128 | .605±.017 | **.670±.069** | .639±.051 | .522±.066 |
| survset_glioma | 37C-index ↑ | .745±.055 | .745±.045 | .673±.066 | **.761±.043** | .500±.000 | .724±.079 | .745±.030 | .701±.115 |
| survset_glioma | 37IBS ↓ | **.153±.016** | – | .225±.050 | .214±.006 | .223±.004 | .181±.029 | .197±.057 | .219±.092 |
| survset_glioma | 37iAUC ↑ | **.865±.008** | .827±.043 | .647±.222 | .833±.021 | .500±.000 | .804±.094 | .825±.041 | .678±.194 |

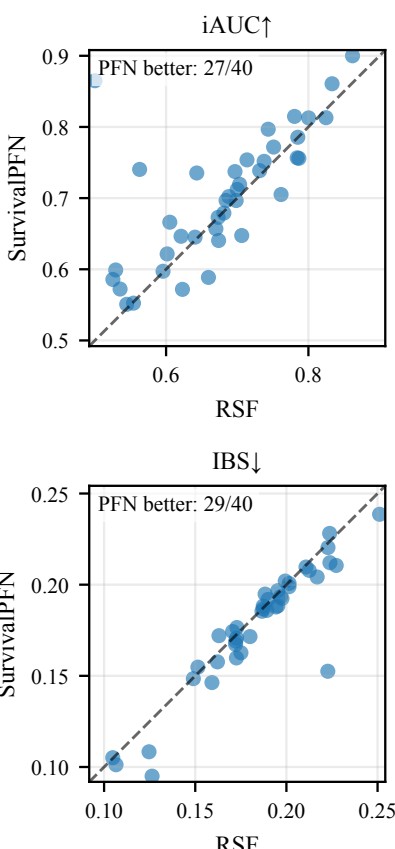

*Figure 5.* Per-dataset SurvivalPFN vs. RSF (mean over splits). Points above the diagonal indicate higher iAUC for SurvivalPFN; points below the diagonal indicate lower IBS.

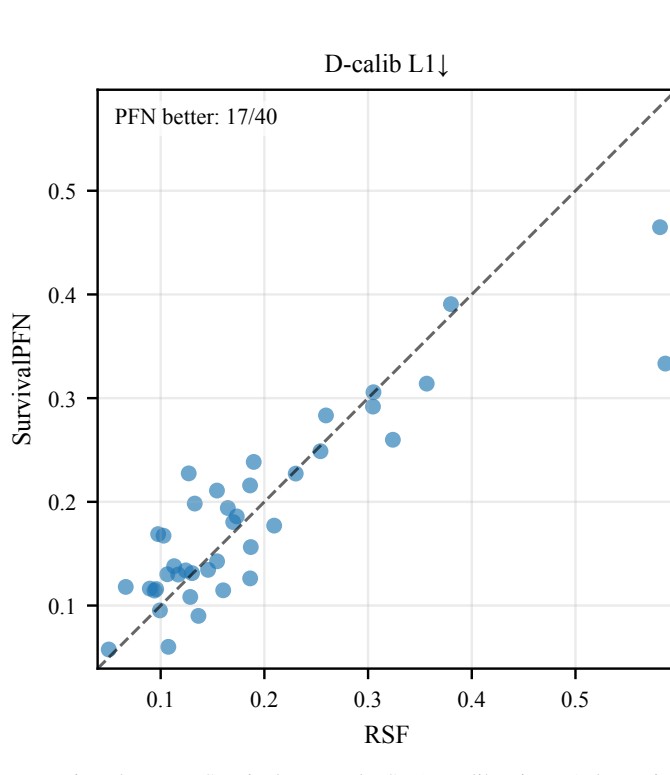

*Figure 6.* Per-dataset calibration comparison between SurvivalPFN and RSF (D-calibration L1; lower is better). Points below the diagonal indicate better calibration for SurvivalPFN.

## A.6. Inference latency and downstream compute

The final SurvivalPFN uses a moderate-size 8-layer, 8-head Transformer with $d_{\text{model}}{=}256$ and $d_{\text{ff}}{=}2048$ (13.2 M parameters) and $J = 64$ time bins. Following prior PFN work, we separate one-time pretraining from per-dataset use: baselines incur per-dataset training plus inference, whereas SurvivalPFN only incurs inference with fixed weights. Baseline training and prediction times are recorded during the benchmark run on the same device as metric evaluation; SurvivalPFN reports inference only. Times are measured as wall-clock around the final model `fit` and the `predict` calls on the preprocessed features, excluding dataset I/O, preprocessing (standardization and one-hot encoding), and metric computation. Predict time includes producing both risk scores and survival curves at the evaluation horizons. For tuned baselines, we additionally record the time spent on hyperparameter selection on the training data. Some classical baselines can also exhibit heavy-tailed runtimes on high-dimensional, ill-conditioned cohorts under the same protocol: for example, CoxPH includes an $\ell_2$ regularization grid-search, and on `survset_aml_bull` the resulting tuning+fit wall-clock can exceed 200 s, inflating mean and standard deviation even though the median per-dataset runtime is far smaller. For neural baselines, predict time can be well below 1 ms per dataset on CPU (e.g., DeepSurv/DeepHit), so rounding to coarse precision may appear as "0"; we report 4-decimal seconds to avoid this ambiguity. SurvivalPFN inference time is heavy-tailed across datasets: the per-dataset mean is 1.53 s but the median is 0.04 s. One dataset with high post-encoding dimensionality (`survset_aml_bull`, $d{=}1675$) takes $\approx 55$ s and dominates the mean; excluding it yields a mean inference time of 0.16 s. The reported pretraining cost for the final model is measured on two RTX 5090 GPUs and is approximately 16.9 hours wall-clock (about 33.8 GPU-hours) for the 1000-epoch run.

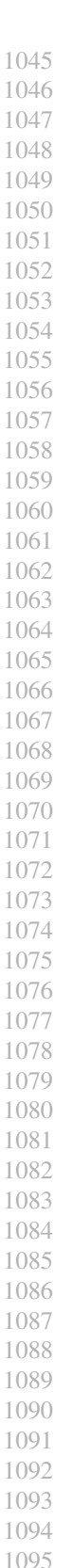

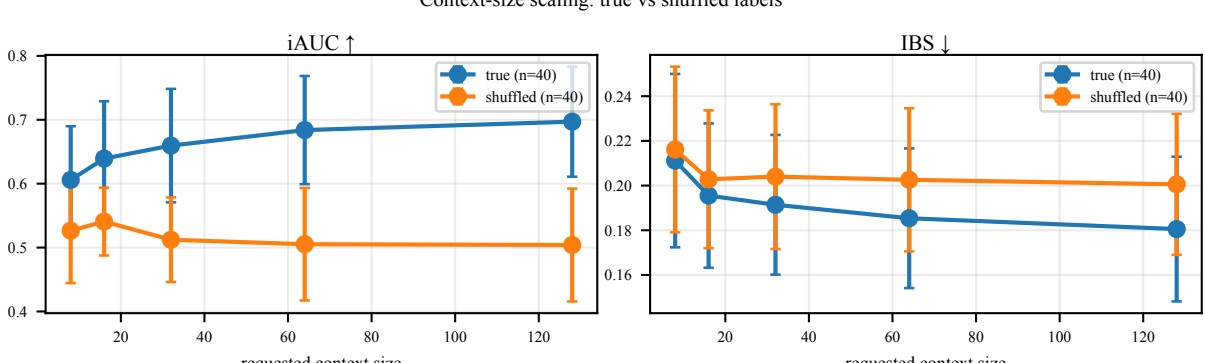

*Figure 7.* Context-size scaling control. We vary the requested context size for SurvivalPFN at inference and compare true labels vs. a label-shuffle control that keeps covariates fixed but permutes the labels $(Z, \delta)$ among context points, breaking the covariate–label correspondence. With correctly paired labels, iAUC improves and IBS decreases with more context. Under the shuffle control, iAUC shows little scaling while IBS improves only mildly and quickly plateaus, consistent with estimating marginal survival rather than covariate-conditioned risk.

### A.7. Synthetic prior specification

The main text (Figure 2) summarizes empirical prior coverage across mechanisms and data statistics. Here we provide the full prior hyperparameters and additional implementation details for reproducibility. For reproducibility, Table 8 lists the full hyperparameters of the survival-task prior used by our final model. Covariates are generated from mixed marginals (Gaussian, uniform, multinomial, Zipf) to form latent causes, which are passed through a random tanh MLP with 3–6 layers and width $2d$ (weights scaled by $1/\sqrt{\text{fan-in}}$, 0.1 weight dropout, Gaussian noise) and then randomly subsampled to $d$ features. With probability $p_{\text{heavy}}$, we apply a heavy-tail transform $x \leftarrow \tanh(x) \cdot \exp(\mathcal{N}(0, 0.5^2))$. A task-dependent fraction of columns is converted to categorical features via one-hot encoding; category logits come from a random linear projection of the continuous source with a fixed logit scale. For event mechanisms, we build an internal representation $x^{\text{rep}}$ that is either linear (identity) or a random MLP with $x_{\text{repr}}$ hidden units and $L_{\text{repr}}$ layers. The covariate score $\eta(x)$ is sampled either from a sparse linear form on $x^{\text{rep}}$ or from a small random tree ensemble (depth, number of trees, and leaf scale in Table 8). Baseline integrated-hazard increments $A_0$ are sampled from a log-random-walk prior, a 1/f Fourier prior, or a Weibull prior and then rescaled to match the target end-of-horizon cumulative hazard. Non-PH effects use a low-rank time-varying term: temporal factors follow a random walk, feature factors are random projections of $x$, and the resulting term is standardized and scaled. AFT tasks use a time warp $H_i(t) = H_0(t \exp(\eta_i))$ with piecewise-constant hazards and linear interpolation of the baseline cumulative hazard on the bin grid. We mix in covariate-dependent censoring (still conditionally independent) with probability 0.05 via a sparse covariate score (sparsity 0.4, coefficient scale 0.4, log-multiplier clipped to $[-6, 6]$) and weakly dependent censoring with probability 0.02 using a Gaussian copula with $\rho \sim \text{Uniform}[0, 0.05]$.

*Table 8.* Synthetic survival-task prior settings (coverage summary + full hyperparameters).

| Hyperparameter | Value |
|---|---|
| **Prior coverage (summary)** | |
| Event families (PH / non-PH / AFT) | 0.45 / 0.25 / 0.30 |
| Event baseline hazards (rw / 1overf / Weibull) | 0.40 / 0.35 / 0.25 |
| Censor baseline hazards (rw / 1overf / Weibull) | 0.40 / 0.35 / 0.25 |
| Covariate score eta(x) | mixture (tree prob=0.40; linear prob=0.60) |
| Raw-time horizon scale (log-uniform) | [0.300, 16.000] |
| Target censoring fraction | p0=0.14; low=0.18 in [0.000, 0.120]; mid=0.36 in [0.010, 0.950]; high=0.32 in [0.900, 0.999] |
| Censoring regimes | covariate-dependent prob=0.05; copula-coupled prob=0.02 (rho in [0.000, 0.050]) |
| Feature types | heavy-tail prob=0.20; categorical fraction¡= 0.40; categorical cardinality in [2, 8] |
| Task quality control | min events=2; min events in context=1; require event in context prob=0.40 |
| **Discretization / time scale** | |
| num_time_bins | 64 |
| time_scale_min | 0.3 |
| time_scale_max | 16.0 |
| **Covariates** | |
| x_gen | scm |
| factor_rank | 4 |
| p_heavy_tail | 0.2 |
| shuffle_features | true |
| p_categorical | 1.0 |
| categorical_max_cols_fraction | 0.4 |
| categorical_cardinality_min | 2 |
| categorical_cardinality_max | 8 |
| categorical_logit_scale | 0.7 |
| x_repr_net | mixture |
| x_repr_p_linear | 0.35 |
| x_repr_hidden | 64 |
| x_repr_activation | relu6 |
| **Event mechanism** | |
| event_mechanism | mixture |
| event_p_ph | 0.45 |
| event_p_aft | 0.3 |
| signal_dist | mixture |
| signal_scale | 2.5 |
| signal_min | 1.5 |
| signal_max | 4.0 |
| event_beta_scale | 2.5 |
| event_beta_sparsity | 0.3 |
| min_signal_abs_corr | 0.05 |
| event_hazard_baseline_kind | mixture |
| event_hazard_end_cum_min | 0.2 |
| event_hazard_end_cum_max | 8.0 |
| event_hazard_end_cum_zero_min | 0.2 |
| event_hazard_end_cum_zero_max | 8.0 |
| event_hazard_end_cum_high_min | 0.2 |
| event_hazard_end_cum_high_max | 8.0 |
| event_hazard_tvc_scale | 2.6 |
| event_hazard_multimodal_prob | 0.25 |
| event_hazard_cure_max | 0.25 |
| event_hazard_cure_zero_max | 0.05 |
| aft_log_time_scale_clamp | 2.0 |
| **Censoring** | |
| censor_target_dist | uniform_mixture |
| censor_min | 0.01 |
| censor_max | 0.95 |
| censor_zero_prob | 0.14 |
| censor_low_prob | 0.18 |
| censor_low_min | 0.0 |
| censor_low_max | 0.12 |
| censor_high_prob | 0.32 |
| censor_high_min | 0.9 |
| censor_high_max | 0.999 |
| censor_copula | gaussian |
| p_dep_censor | 0.02 |
| copula_rho_min | 0.0 |
| copula_rho_max | 0.05 |
| p_informative_censor | 0.05 |
| cens_x_sparsity | 0.4 |
| cens_x_scale | 0.4 |
| cens_hazard_baseline_kind | mixture |
| cens_hazard_end_cum_min | 1.0 |
| cens_hazard_end_cum_max | 4.0 |
| censor_rate_min | 0.0 |
| censor_rate_max | 0.999 |
| **Task-quality control** | |
| min_events | 2 |
| min_events_in_context | 1 |
| require_event_in_context_prob | 0.4 |

