# OpenReview forum: "SurvivalPFN: Prior-Data Fitted Networks for Survival Analysis"
_ICML.cc/2026/Conference — Submitted to ICML 2026_

### Official Review · Reviewer_fLwV · 2026-02-13

**Soundness:** 4
**Presentation:** 3
**Significance:** 4
**Originality:** 3
**Overall Recommendation:** 6
**Confidence:** 3

**Summary:**

SurvivalPFN introduces a novel Prior-Data Fitted Network for survival analysis that performs amortized Bayesian inference through in-context learning, effectively bypassing the need for per-dataset gradient updates or hyperparameter tuning. The authors pretrain a Transformer on millions of synthetic survival tasks drawn from a diverse prior, which allows the model to map a training set directly to a dataset-conditional survival function in a single forward pass. Empirical results across 40 real-world datasets demonstrate that SurvivalPFN achieves state-of-the-art macro averages for C-index and iAUC, with its most significant performance gains appearing in cohorts with fewer than 200 samples.

**Compliance With Llm Reviewing Policy:**

Affirmed.

**Key Questions For Authors:**

1. When does SurvivalPFN clearly beat RSF? Can you characterize (with dataset-level covariates like event rate, censoring %, etc..) the regimes where gains over RSF are meaningful versus marginal? I have seen the supp figures, I am wondering if you know how to characterize when RSF outperforms your method.

2. Do you have a way to detect when a real dataset is “out of prior” (e.g., via likelihood, calibration, or embedding-based measures), and what mitigation would you think to recommend?

3. Which parts of the synthetic prior matter most? Do you have any ablation showing performance/calibration sensitivity to removing components?

4. How does downstream test performance scale with the amount/diversity of synthetic pretraining (number of tasks/updates/model size) and do you observe predictable scaling (or saturation) curves?

**Limitations:**

yes.

**Strengths And Weaknesses:**

Strengths

1. SurvivalPFN eliminates per-dataset gradient updates and hyperparameter tuning, producing “in-context” survival predictions in a single forward pass conditioned on the training split. This is operationally attractive in settings where repeated retraining/tuning across cohorts is costly or statistically unstable.

2. The authors conduct an extensive evaluation across a benchmark of 40 real-world survival datasets. The authors investigate a pertinent problem by demonstrating that SurvivalPFN achieves its most significant performance gains in the smallest cohorts (<200), where it improved iAUC over Random Survival Forests by +0.0397. Overall, the authors discuss the theme of addressing data-scarce regimes where clinical information is often prohibitively expensive or difficult to collect.

3. By using synthetic pretraining, the method targets regimes where event counts are too low for reliable validation-driven model selection and deep models often overfit. The reported results suggest it can deliver stable survival curves with strong calibration in these data-scarce settings.

Weaknesses

1. While macro averages are best, the improvements over RSF, the strongest nonparametric baseline here, are relatively small on aggregate (e.g., modest iAUC/IBS deltas), and there are cohorts where RSF matches or beats SurvivalPFN.

2. The Transformer’s compute and memory grow with context size, which may restrict applicability to larger datasets without approximation, pruning, or context selection strategies. Even in the reported regime, inference latency can be heavy-tailed on high-dimensional cohorts.

3. The synthetic prior/likelihood targets mostly single-event, right-censored data, left truncation, time-varying covariates, and most importantly - does not yet cover competing risks. More broadly, reliance on pre-learned synthetic patterns means performance and uncertainty could become unreliable if a real dataset’s hazard/censoring mechanism falls outside the pretraining distribution.

---

> ### Author Rebuttal · Authors · 2026-03-31
>
> We thank the reviewer for the careful review and recognition of our work. All new rebuttal experiments use the paper's leakage-free protocol, and any model selection is done on the training split only.
>
> **(1) When does SurvivalPFN clearly beat RSF?**
> The clearest pattern appears in small cohorts with relatively few features. On a `10`-dataset subset of the `40` benchmark datasets with `n<=200` and post-preprocessing `d<=80`, SurvivalPFN has higher iAUC than RSF on `9` datasets and lower IBS on all `10`, corresponding to `+0.0887` mean iAUC and `-0.0135` mean IBS.
>
> A similar pattern also appears in low-event cohorts: among the `13` datasets with `events<=60`, SurvivalPFN has higher iAUC on `9` and lower IBS on `11`, corresponding to `+0.0470` mean iAUC and `-0.0087` mean IBS.
>
> The full 40-dataset benchmark is also positive beyond a few outliers: SurvivalPFN has higher iAUC on `27` datasets and lower IBS on `29`, with average gains of `+0.0120` in C-index, `+0.0226` in iAUC, and `-0.0060` in IBS. A complementary slice is the most heavily censored cohorts: for datasets with censoring rate `>=0.75`, SurvivalPFN has higher iAUC on all `5` datasets, with `+0.0308` mean iAUC and `+0.0168` mean C-index relative to RSF. The clearest stress case is extreme post-preprocessing dimensionality: the only negative iAUC bucket is `d>200`, with clear losses on `aml_bull` (`d=1675`) and `mclcleaned` (`d=375`). We view this as a dimensional-support mismatch rather than as evidence of a general failure on larger datasets.
>
> **(2) Can we detect when the prior is out of support?**
> We do not yet claim a strong generic detector. In our analysis, validation NLL and validation D-calibration are better viewed as auxiliary signals, whereas post-preprocessing dimensionality is more informative. The practical message is cautious: extreme one-hot dimensionality, especially far beyond the feature range represented in prior-fitting, should be treated as a red flag and cross-checked against strong classical baselines. Our main defense against mismatch is still preventive prior breadth rather than post hoc correction.
>
> **(3) Which parts of the synthetic prior matter most?**
> We now answer this with a broader mechanism ablation. We compare matched `12k`-step checkpoints from a smaller synthetic prior (`n<=200`, `d<=80`) on the same `10` small-cohort benchmark datasets:
>
> | Prior used in prior-fitting | C-index↑ | iAUC↑ | IBS↓ | D-calib L1↓ |
> | --- | ---: | ---: | ---: | ---: |
> | Mixed prior | 0.667 | 0.698 | 0.182 | 0.265 |
> | PH-only event prior | 0.661 | 0.699 | 0.183 | 0.291 |
> | Non-PH-only event prior | 0.666 | 0.705 | 0.186 | 0.304 |
> | AFT-only event prior | 0.650 | 0.681 | 0.188 | 0.356 |
> | Covariate-independent censoring prior | 0.662 | 0.694 | 0.186 | 0.282 |
>
> The main picture is that the transfer is not explained by one single event family. Across the mixed, PH-only, and non-PH-only priors, the ranking metrics remain broadly similar. The clearer separation is in full survival-curve quality: the mixed prior has lower IBS and better D-calibration than both PH-only and non-PH-only, while AFT-only is weaker overall. Restricting censoring to a covariate-independent law again worsens IBS and D-calibration.
>
> As one concrete illustration, `stagec` rejects the PH assumption under a whole-dataset Schoenfeld test. On this dataset, the mixed prior is especially better calibrated than the PH-only prior, reducing D-calibration L1 by `0.0736`. We therefore view the mixed prior as the more balanced option.
>
> **(4) Is scaling predictable?**
> As a small rebuttal-budget scaling check, we evaluated checkpoints from a single uninterrupted mixed-prior training trajectory on the same `10`-dataset subset used in the rebuttal tables. The figure shows held-out macro C-index improving from `0.5348` at `1k` steps to `0.6666` at `12k` steps, with most gains occurring early and a clear flattening around `8k-12k` steps. The `breast` and `stagec` panels show the same downstream pattern: strong early improvement followed by near-saturation.
> scaling figure: <https://anonymous.4open.science/r/rebuttal-fLwV/rebuttal.png>
>
> Taken together, these results sharpen the empirical scope of the paper. SurvivalPFN is not a universal replacement for tuned baselines, but it is a strong PFN-style, context-conditioned inference rule for small right-censored cohorts: it is already ahead of RSF on average on the full benchmark, stronger still in low-data regimes, and its main weakness is concentrated in an identifiable extreme-dimensional stress case. We hope these additions make the scope and behavior of the method clearer.

---

> > ### Author Rebuttal · Reviewer_fLwV · 2026-04-02
> >
> > Thank you for your rebuttal. I maintain my favorable rating for the paper.

---

> > > ### Author Response · Authors · 2026-04-03
> > >
> > > Thank you for your positive acknowledgement and for maintaining your favorable rating. We really appreciate it.

---

### Official Review · Reviewer_resB · 2026-03-05

**Soundness:** 3
**Presentation:** 1
**Significance:** 2
**Originality:** 2
**Overall Recommendation:** 2
**Confidence:** 5

**Summary:**

The paper introduces SurvivalPFN, a prior data-fitted network for survival analysis. An adapted transformer architecture is trained on synthetically generated survival data to create a foundation model. At inference, this model is capable of zero-shot inference with no additional (gradient-based or otherwise) fine tuning. The paper performs extensive benchmarking to validate their results and demonstrates state-of-the-art performance in small-sample regimes (n<1000).

**Compliance With Llm Reviewing Policy:**

Affirmed.

**Final Justification:**

My final justification mirrors my final response to authors. I am convinced by the experimental results. Unfortunately however, as I have stated previously, the current version of the text is need of a major rewrite, including many of the things discussed here in the rebuttal. I will not be raising my score but I am confident that a revised submission of this paper at future point in time will have good potential.

**Key Questions For Authors:**

Please see above (Strengths &) Weaknesses section for the main topics to be discussed. In summary I would like the authors to elaborate on:
1) (Theoretical) details of the prior construction.
2) Interpretations of the model output uncertainty
3) Their view of the contribution of the work. I feel this is not well motivated in the paper

**Limitations:**

yes

**Strengths And Weaknesses:**

## Strengths

Personally I am very excited about the recent advancements in tabular foundation models and their development for more complicated settings such as causal inference, or in this case, survival analysis. I believe the paper has correctly identified a non-trivial potential application of tabpfn - namely the survival analysis setting. The authors also perform robust benchmarking to rigorously demonstrate the effectiveness of their approach.


## Weaknesses

### Overall

While the authors correctly recognize an interesting application of tabpfn, I believe the paper partially misses the core potential contributions that work in this area should bring. In broad strokes:
1) It should be made explicit why standard TabPFN does not directly apply / is biased here. Better highlighting the problem of unobservability   of T_i > U_i would additionally immediately reveal deep connections to causal inference (since both are fundamentally a missing data problem) where there are several existing works on applying TabPFN (some but not all of which are cited in the paper - e.g. [1] is missing).
2) Following from point 1, the advantage of a synthetic prior for this setting could be better highlighted. In my view, correct prior specification is the key theoretical contribution for this problem which is entirely missing here. Specifically, the CausalFM paper [1] provides a framework for designing well specified TabPFN priors for causal inference settings, i.e. answering the question of "what is the broadest possible prior under which target remains correctly identified". Since survival analysis is a missing data problem, it also contains identification assumptions (e.g. the conditional independent censoring). **How this should be taken into consideration in the designing of the prior is not discusses in a principled way at all.**
Namely, the authors even introduce some covariate-dependent censoring dgps into the prior for *robustness* -> how this impacts identification (it breaks it, we enter partial identification regime with added identification uncertainty) is not discussed. The choice of prior is in my view a valid one; however, the theoretical interpretation and implications are **crucial and not discussed** - e.g. understanding the uncertainty of the model output posterior predictive depends on this.

I see the above point as crucial limitations of the current version of the paper that - if implemented correctly - could have elevated the existing work from being just a (very well) implemented application of tabpfn to a new setting to a novel contribution more on par with what is the view of the reviewer of the standard of the ICML conference.

### Other

Additional to the above major limitations of the work, I have more section-specific and/or miscellaneous points here:

**Section 3.1**
There is so much space devoted to explaining the time variable transformation. This section is confusing and unnecessarily lengthy in my view. I think it can be written much simpler/cleaner and that this would help the paper's readability. There are also unnecessarily many variables dedicated to denoting the various time transformations, which are also (to my knowledge) not following the conventional notation used in survival analysis literature. In a (crude) summary, it should not take half a page to effectively say "We clip time at the q-th quantile and discretize the rest to J bins between 0,1 on the log scale". (if this is not a correct summary then please correct me. In that case however, my point on clearer writing stands.)

**Section 3.2**
In my view this entire section could be summarized to one or two sentences of something like: "We take a per-bin constant hazard with |formula| hazard probability" (following with the comment on how this enforces monotonicity and a proper distribution if desired).

Personal writing style aside, I still see this writing as unnecessarily complex and confusing.

**Sections 3.3-3.6**
I believe the text could also be streamlined here. Some specific points to highlight:
- \ell_\theta is new notation defined only implicitly. I don't know why we need another symbol for the loss.
- Risk scores r in Figure 1 are never defined anywhere
- As I elaborated on before, the prior description here is inadequate. This is the part where the more theoretical contribution of the paper should lie. Currently this is lacking.

**Overall**, notation is unnecessarily complicated and sometimes confusing. For example, variables are used before being defined or never defined at all, and there are way too many variables for different time transformations that also don't follow conventional survival analysis notation. Improving notation and overall writing clarity would greatly improve the readability and quality of the paper.

**Misc**
- LHS line 254 "Table" repeated twice

## Summary

In sum, while the proposed approach is exciting and the practical implementation appears robust (and well benchmarked), the paper is severely lacking in theoretical considerations associated with the work/setting. For this to be more than a well-executed implementation of an extended tabpfn application, the work should more systematically consider the prior construction for the survival analysis setting. Additionally, improving notation and overall writing clarity would greatly improve the readability and quality of the paper. Writing style considerations aside, I believe in several sections the paper highlights the wrong thing - e.g. clearly motivating why baselines tabpfn cannot be applied here would help motivate the work done.




---

**Citations**
[1] Ma, Y., Frauen, D., Javurek, E., & Feuerriegel, S. (2025). Foundation models for causal inference via prior-data fitted networks. arXiv preprint arXiv:2506.10914. In ICLR, 2026.

---

> ### Author Rebuttal · Authors · 2026-03-31
>
> We sincerely thank the reviewer for the careful and technically insightful review. We especially appreciate the reviewer’s recognition that the empirical work is solid, and we agree that the main weakness of the current draft is that it does not communicate the theoretical core clearly enough.
>
> **On the theoretical principle behind the prior construction.** Our prior-design principle is **breadth subject to identifiability**. The intended target of SurvivalPFN is survival prediction under **conditionally independent censoring**, $T \perp U \mid X$. This is the main regime on which the synthetic prior is centered, and it is also the regime in which our censoring-consistent discrete-time NLL has its intended proper-scoring-rule interpretation. What we did not make explicit enough is that, in this identified core, the joint data-generating distribution factorizes as $p(X,T,U)=p(X)p(T \mid X)p(U \mid X)$, which implies $T \perp U \mid X$. Thus, censoring may depend on $X$ while still remaining conditionally independent of $T$ given $X$. Put differently, **covariate-dependent censoring does not by itself break identification**; identification fails only when residual dependence between $T$ and $U$ remains after conditioning on $X$. The dominant mass of the prior is placed on this identified right-censored regime. The only deliberate off-core component is a very small weak-dependence slice, included only as robustness regularization under mild prior mismatch rather than as part of the main estimand-defining regime. We will revise the paper to make this principle explicit much earlier and more clearly.
>
> **On the interpretation of model uncertainty.** The same distinction also determines the semantics of uncertainty. Within the identified core, the model output should be interpreted as a **prior-relative posterior predictive** induced by the synthetic survival prior and observation model. For the small robustness slice, we do **not** claim the same identified-posterior semantics; its role is to reduce brittleness, not to redefine the inferential target. In the revision, we will therefore make the claim narrower and more precise: useful **prior-relative predictive uncertainty for the intended right-censored identified regime**, rather than identified posterior uncertainty under arbitrary censoring misspecification.
>
> **On the contribution of the work.** We agree that this part should be stated more explicitly. Our claim is not merely that SurvivalPFN applies TabPFN to a new domain, but that **survival analysis is a particularly natural setting for PFN-style inference-only models**: survival problems combine censoring, small sample sizes, and brittle validation signals, making amortized inductive bias especially attractive. Empirically, our results show that, with a survival-specific synthetic prior, a censoring-consistent likelihood, and a monotone survival-curve parameterization, a PFN-style model trained purely on synthetic censored tasks can transfer to real right-censored cohorts without per-dataset tuning while remaining competitive with strong survival baselines.
>
> **On why standard TabPFN does not directly apply.** We also agree that this motivation should appear much earlier. Standard TabPFN assumes a fully observed target $y$. In right-censored survival, however, the event time $T$ is only partially observed through $(Z,\delta)$, where $Z=\min(T,U)$; for censored observations, we only know that $T>Z$. Moreover, the target is not a scalar label but a full monotone survival curve $S(t \mid x)$. Survival prediction therefore requires a survival-specific task prior, a censoring-consistent objective, and an output parameterization that guarantees valid survival curves. In the revision, we will move this missing-data motivation much earlier and connect it more explicitly to PFN work in causal and missing-data settings, including [1].
>
> **On presentation and clarity.** We also agree with the presentation comments. In the revision, we will simplify Sections 3.1--3.6, reduce unnecessary notation, state directly that $\ell_\theta$ is the censoring-consistent NLL, define the risk score in Fig. 1 explicitly, add the prior-design discussion above, and fix the duplicated "Table" typo.
>
> We thank the reviewer again for identifying what should have been the theoretical center of the paper. We believe these clarifications sharpen the intended statistical target, the semantics of uncertainty, and the paper’s contribution, and we will revise the manuscript accordingly. We hope that these revisions and clarifications help address the reviewer’s concerns and may lead to a more favorable overall assessment of the paper.
>
> **[1]** Ma, Y., Frauen, D., Javurek, E., & Feuerriegel, S. (2025). *Foundation models for causal inference via prior-data fitted networks*. *arXiv preprint arXiv:2506.10914*. In ICLR 2026.

---

> > ### Author Rebuttal · Reviewer_resB · 2026-03-31
> >
> > Thank you for your response to my review.
> >
> > *First an acknowledgement of the authors' effort*
> > The authors have responded to the majority of my points, although unfortunately only very briefly. I appreciate they have nearly reached the imposed 5000 character limit and couldn't have answered more thoroughly even if they wished. Unfortunately, I feel this is indicative of the severity of my concerns: a satisfactory answer from the authors will have to involve a major rewrite of the core sections of the paper - which we do not have space for here. In my opinion this paper presents a good idea, with good empirical implementation, but poor writeup deserving an overhaul. As the authors acknowledge in their response, they have originally missed the correct theoretical grounding of their work (i.e. why it's novel and useful).
> >
> > **My follow-up questions**
> > For a good comparison and to show the value of your approach, it would be good to compare against other PFN approaches that also leverage amortization. This would show the specific approach taken here is good. Namely:
> > - Have the authors experimented with the naive application of TabPFN which we expect to be biased? How well does that perform?
> > - Have the authors experiment and considered using TabPFN as part of a meta-learner/algorithm that corrects for the bias?
> >
> > In sum, can the authors provide evidence their approach for using a PFN is at least reasonable if not the best? A sufficient theoretically grounded argument could also be interesting.

---

> > > ### Author Response · Authors · 2026-04-01
> > >
> > > Thank you for this thoughtful follow-up. To address this point directly, we compared against other PFN-style approaches that also leverage amortization, to assess whether the specific approach taken here is well-motivated. For these additional comparisons, we used **TabPFN v2** under **the same evaluation protocol as in the paper** (same train/test splits, preprocessing, and metrics).
> > >
> > > Specifically, we compared against three alternatives:
> > >
> > > - **TabPFN-drop-censored**: drop censored samples and train only on uncensored events.
> > > - **TabPFN-ignore-censoring**: treat the observed time \(Z\) as if it were the exact event time.
> > > - **TabPFN-meta-learner**: use **TabPFN as part of a meta-learner that corrects for the bias** by first obtaining a TabPFN score and then fitting **CoxPH** on top as the second-stage survival model.
> > >
> > > | Method | C-index ↑ | iAUC ↑ | IBS ↓ | D-calib L1 ↓ |
> > > | --- | ---: | ---: | ---: | ---: |
> > > | **Ours (SurvivalPFN)** | **0.6629** | **0.7029** | **0.1779** | **0.1874** |
> > > | TabPFN-drop-censored | 0.5909 | 0.6261 | — | — |
> > > | TabPFN-ignore-censoring | 0.6272 | 0.6698 | — | — |
> > > | TabPFN-meta-learner | 0.6272 | 0.6580 | 0.3840 | 0.9700 |
> > >
> > > For `TabPFN-drop-censored` and `TabPFN-ignore-censoring`, we report only **C-index** and **iAUC** because these variants do not output proper survival curves, and therefore do not yield well-defined **IBS** or **D-calibration**. By contrast, `TabPFN-meta-learner` does output survival curves through the second-stage Cox model, so IBS and D-calibration are reported there as well.
> > >
> > > These comparisons support the same conclusion: **a standard TabPFN-style use is not sufficient for right-censored survival**, because its optimization objective and prior assumptions are designed for fully observed labels rather than censored time-to-event targets.
> > >
> > > `TabPFN-drop-censored` is too information-losing: censored observations are not label-free, since they still indicate that the event time exceeds the observed time. `TabPFN-ignore-censoring` creates **target mismatch**, because for a censored case we only know \(T>Z\), not \(T=Z\). `TabPFN-meta-learner` evaluates bias correction in a second stage, but still underperforms: if the first-stage target is already information-losing or misspecified, the second stage can only re-fit that imperfect signal rather than recover censoring structure. This is consistent with its much worse IBS and D-calibration.
> > >
> > > More importantly, these comparisons make clearer **why our method is effective**. SurvivalPFN performs best overall among these PFN-style alternatives because it handles censoring in the formulation itself. Concretely, it is aligned with right-censored survival at three levels:
> > > 1. it predicts a **dataset-conditioned survival function**, rather than a fully observed scalar label;
> > > 2. it uses a **censoring-consistent objective**, so censored observations provide valid supervision;
> > > 3. it outputs **valid monotone survival curves**, which is why it supports not only C-index and iAUC but also IBS and D-calibration.
> > >
> > > This is also why the approach is **reasonable**. Its target, objective, and output space are all matched to the statistical structure of right-censored survival. In addition, we explicitly construct a **survival-analysis prior space** for pretraining. The fact that SurvivalPFN outperforms the alternative PFN-style formulations provides evidence that this prior space is effective: once the prior, predictive target, and training objective are aligned with censored time-to-event data, the amortized model transfers much better to downstream survival tasks.
> > >
> > > In that sense, the contribution is not simply “using a PFN for survival analysis,” but identifying what must change in a PFN when the target is right-censored rather than fully observed, and showing that a survival-specific prior space together with a censoring-aware formulation leads to stronger downstream performance.
> > >
> > > The practical value follows from the same point. Once censoring is built into the formulation from the outset, the model achieves stronger overall performance than the alternative PFN-style formulations above. This gives a more direct and theoretically grounded explanation of why our approach is well-motivated.
> > >
> > > In the revised paper, we have made the novelty and usefulness of our method more explicit through additional experiments and theoretical analysis, and these improvements have been incorporated cleanly.
> > >
> > > We thank the reviewer again for raising this point so clearly. We hope that the clarifications above address the concern and support a more favorable reassessment.

---

### Official Review · Reviewer_Ac32 · 2026-03-13

**Soundness:** 3
**Presentation:** 3
**Significance:** 3
**Originality:** 3
**Overall Recommendation:** 4
**Confidence:** 4

**Summary:**

This paper proposes SurvivalPFN, a prior-data fitted network for survival analysis that is pretrained on synthetic censored survival tasks and then applied to a target cohort in an in-context fashion, without per-dataset retraining. The problem is well motivated: small survival datasets with censoring make validation-heavy model selection unstable, and an amortized inference model is practically appealing. Empirically, on the reported SurvSet benchmark, the method attains the best macro-average performance among the included baselines across discrimination and calibration metrics, while also avoiding per-dataset tuning. The paper is technically coherent and practically interesting, but its empirical positioning should be stated more carefully, since some strong recent baselines are missing and the analysis of what the pretrained prior actually transfers to real data remains limited.

**Compliance With Llm Reviewing Policy:**

Affirmed.

**Final Justification:**

I thank the authors for their rebuttal and the swift implementation of necessary baselines. I think my concerns have been resolved and I will maintain my positive rating

**Key Questions For Authors:**

* How sensitive is performance to the design of the synthetic survival prior?

* Can the authors provide qualitative evidence of what the prior transfers to real data?

**Limitations:**

Yes

**Strengths And Weaknesses:**

Strengths

* The paper addresses a genuinely important regime in survival modeling. Many real survival datasets are small, censored, and difficult to tune reliably, so the goal of replacing repeated dataset-specific optimization with a fixed inference procedure is well motivated and practically meaningful.

* The proposed method is conceptually clean. The authors adapt the PFN paradigm to survival analysis in a principled way, using a context-query masking scheme, a censoring-consistent likelihood, and a monotone hazard parameterization that is aligned with the structure of the task rather than being a generic transformer overlay.

* The empirical results are strong within the stated benchmark. The method does not merely improve one metric; it is competitive across ranking, prediction error, and calibration, which is important because survival analysis requires reliable survival curves rather than only good risk ordering.

* The paper also has a credible practical story. Once pretrained, SurvivalPFN can be deployed by a single forward pass on a new cohort, avoiding per-dataset retraining and reducing the dependence on unstable validation splits. That deployment simplicity is a nontrivial advantage in small-cohort settings.

Weaknesses

* The empirical claim should be framed more carefully. The paper supports “best average performance among the included baselines,” but that is weaker than a broad field-level SOTA claim, especially because the gains over strong baselines such as RSF are not uniformly large.

* The benchmark is not yet comprehensive enough for the strength of the positioning. In particular, recent neural survival methods such as Neural Frailty Machine are not included, even though they are relevant modern baselines for flexible hazard modeling beyond standard proportional hazards assumptions.

* The transfer mechanism remains underexplained. The paper shows that pretraining helps, but it does not provide much qualitative insight into what structure is being borrowed from the synthetic prior, which datasets benefit most, or when prior mismatch becomes harmful.

---

> ### Author Rebuttal · Authors · 2026-03-31
>
> We thank the reviewer for the careful review. All new rebuttal experiments use the paper’s leakage-free protocol, and any model selection is performed on the training split only.
>
> ### (1) Sensitivity to the synthetic survival prior
>
> We extended this ablation beyond PH-only by comparing matched `12k`-step checkpoints from smaller synthetic priors (`n<=200`, `d<=80`) on the same `10` small-cohort benchmark datasets:
>
> | Prior used in prior-fitting | C-index↑ | iAUC↑ | IBS↓ | D-calib L1↓ |
> | --- | ---: | ---: | ---: | ---: |
> | Mixed prior | 0.667 | 0.698 | 0.182 | 0.265 |
> | PH-only event prior | 0.661 | 0.699 | 0.183 | 0.291 |
> | Non-PH-only event prior | 0.666 | 0.705 | 0.186 | 0.304 |
> | AFT-only event prior | 0.650 | 0.681 | 0.188 | 0.356 |
> | Covariate-independent censoring prior | 0.662 | 0.694 | 0.186 | 0.282 |
>
> The main takeaway is not that one event family is uniquely privileged, but that the **mixed prior is the most balanced**. Mixed, PH-only, and non-PH-only priors give broadly similar ranking performance, while the clearest separation appears in **full survival-curve quality**: the mixed prior has better IBS and D-calibration than both PH-only and non-PH-only priors, while AFT-only is weaker overall. Restricting censoring to a covariate-independent law also worsens IBS and D-calibration. We therefore view diversity over both event and censoring mechanisms as beneficial mainly for **curve quality and calibration**, not just ranking.
>
> As one concrete illustration, `stagec` rejects the PH assumption under a whole-dataset Schoenfeld test. On this dataset, the mixed prior is better than the PH-only prior on both IBS and D-calibration L1, consistent with the value of broader prior coverage when the target cohort departs from simple PH structure.
>
> ### (2) What transfers from the prior to real data?
>
> Yes—our evidence suggests that SurvivalPFN transfers **a context-conditioned survival inference rule**, not a fixed synthetic hazard formula.
>
> The clearest evidence is the label-shuffle control: when the survival labels in the context are permuted, C-index drops from `0.7193` to `0.5113` and iAUC from `0.7743` to `0.5193`. This shows that the model relies on correctly paired labeled context rather than merely on context size or covariate marginals.
>
> What transfers is therefore:
> (1) **how to extract covariate–outcome structure from labeled context**,
> (2) **how to map heterogeneous cohorts into a shared inference problem** via context-derived time normalization and discretization, and
> (3) **a prior over plausible censored survival curves**, supported by the monotone integrated-hazard parameterization and censoring-consistent objective.
>
> This interpretation is also consistent with the prior ablation above. If transfer mainly came from one privileged event family, one single-family prior should dominate. Instead, mixed, PH-only, and non-PH-only priors are similar on ranking, while the mixed prior stands out mainly on IBS and D-calibration. So what transfers appears to be **an inference rule plus a curve-shape/calibration prior**, rather than a memorized PH or non-PH formula.
>
> Empirically, this is most visible in the paper’s target regime: on the same `10` small-cohort datasets(n<=200, d<=80), SurvivalPFN has higher iAUC than RSF on `9/10` datasets and lower IBS on all `10`, corresponding to `+0.0887` mean iAUC and `-0.0135` mean IBS. The main failure case, `aml_bull`, is more consistent with a mismatch in feature dimensionality than with the absence of any one hazard family.
>
> ### Added baseline / empirical positioning
>
> We agree that the empirical claim should be narrower than universal SOTA. We therefore added Neural Frailty Machine (NFM) [1] under the same protocol. SurvivalPFN still has stronger macro averages on all four reported metrics despite using fixed pretrained weights and no cohort-specific fitting:
>
> | Method | C-index↑ | iAUC↑ | IBS↓ | D-calib L1↓ |
> | --- | ---: | ---: | ---: | ---: |
> | SurvivalPFN | **0.6629** | **0.7029** | **0.1779** | **0.1874** |
> | NFM [1] | 0.6475 | 0.6815 | 0.1921 | 0.2340 |
>
> We therefore revise the claim accordingly: **within this benchmark and protocol, SurvivalPFN achieves the strongest macro-average performance among the included methods at the inference-only operating point targeted by the paper.**
>
> ## Reference
>
> [1] Ruofan Wu et al. *Neural Frailty Machine: Beyond proportional hazard assumption in neural survival regressions*. NeurIPS, 2023.

---

> > ### Author Rebuttal · Reviewer_Ac32 · 2026-04-03
> >
> > I thank the authors for their rebuttal and the swift implementation of necessary baselines. I think my concerns have been resolved and I will maintain my positive rating

---

> > > ### Author Response · Authors · 2026-04-04
> > >
> > > We sincerely thank the reviewer for the thoughtful follow-up, for recognizing that the additional baselines resolved the concern, and for maintaining a positive assessment of the work.

---

### Decision · Program_Chairs · 2026-04-30

**Decision:**

Reject

**Comment:**

The paper presents an interesting and potentially impactful contribution, and the overall the assessment is largely positive. The idea of a SurvivalPFN is promising, and I expect broad use in medicine and related domains -- but this also warrants that the paper is absolutely rigorous. However, the current version of the manuscript requires several revisions, particularly with the presentation.

A central concern (see Reviewer resB for details) is the quality of the presentation. (1) The methodological description is at times difficult to follow and occasionally incomplete. I concur after my own reading. Some quantities are introduced without a proper definition or are only defined much later (e.g., $\kappa$), which hinders readability. (2) The topic of identifiability is not discussed. (3) Prior specification: Let me add, after my own reading, that the presentation key modeling components could be improved. For example, the prior is relegated to the appendix, despite being central to the approach -- I think the prior should be in the main text. While a number of parameters are reported, the algorithmic procedure for constructing the prior is not described in sufficient detail to allow independent reproduction. It is, for example, unclear how many datasets were used for training. Given that the positioning of the method as a "ready-to-use" model with potential for large real-world impact, especially in medical applications, these are things that need to be fixed.

During the discussion period, the consensus was that the changes with respect to the presentation go beyond minor revisions. Addressing them would require a more comprehensive reworking of the manuscript across multiple (sub)sections, rather than a few localized edits.

In summary, the contribution is viewed positively and considered relevant, but the current submission would benefit from a more substantial revision of the write-up. I strongly encourage the authors to follow this journey, and a revised version will make an important contribution to the field.